# Light-driven modulation of proximity-enhanced functionalities in hybrid nano-scale systems

Mattia Benini [1,2] ✉, Umut Parlak[2], Sophie Bork[2], Jaka Strohsack[3], Richard Leven[2], David Gutnikov[2], Fabian Mertens[2], Evgeny Zhukov [2], Rajib Kumar Rakshit[1], Ilaria Bergenti [1], Andrea Droghetti[4], Andrei Shumilin[3,5], Tomaz Mertelj [3], Valentin Alek Dediu [1] ✉ & Mirko Cinchetti [2] ✉

Advancing quantum information and communication technology requires smaller and faster components with actively controllable functionalities. This work presents an all-optical strategy for dynamically modulating magnetic properties via proximity effects controlled by light. We demonstrate this concept using hybrid nanoscale systems composed of $C_{60}$ molecules prox-imitized to a cobalt metallic ferromagnetic surface, where proximity interactions are particularly strong. Our findings show that by inducing excitons in the $C_{60}$ molecules with resonant ultrashort light pulses, we can significantly modify the interaction at the Cobalt/$C_{60}$ interface, leading to a remarkable 60% transient shift in the frequency of the Co dipolar ferromagnetic resonance mode. This effect, detected via a specifically designed time-resolved Magneto-Optical Kerr Effect (tr-MOKE) experiment, persists on a timescale of hundreds of picoseconds. Since this frequency shift directly correlates with a transient change in the anisotropy field—an essential parameter for technological applications—our findings establish a new material platform for ultrafast optical control of magnetism at the nanoscale.

The quest to manipulate magnetism without external magnetic fields is driving innovations in information and memory storage devices. A particularly promising approach involves using femtosecond laser pulses to either quench[1,2] or switch[3] magnetization. This interaction between ultrashort laser pulses and magnetically ordered materials has spurred the dynamic field of femto-magnetism[4]. Achieving optical control of magnetism with sub-wavelength spatial resolution could introduce new functionalities, such as optical spin-switching for information recording at femtosecond speeds[5,6]. While ultrafast optical control has been demonstrated in dielectric materials[7], coherent control on the femtosecond time scale remains challenging in metallic

systems due to rapid electron-hole pair dephasing and coherence loss caused by Coulomb screening[8].

To overcome these limitations, hybrid heterostructures combining molecular systems with metallic ferromagnets have emerged as a promising platform[9,10]. Proximity effects at such molecular-metal interfaces can induce substantial modifications of the magnetic properties, including enhanced anisotropy[11–14], increased coercivity[15,16] and modified domain structures. Recent studies have revealed that molecular adsorption can result in strong hybridization between the molecular orbitals and the metallic surface states, giving rise to a Correlated Ferromagnetic Glass (CFG) state[17]. In this state, the inter-

[1]ISMN-CNR, Via Piero Gobetti 101, 40129 Bologna, Italy. [2]TU Dortmund University, Otto-Hahn-Straße 4, 44227 Dortmund, Germany. [3]Jozef Stefan Institute, Jamova Cesta 39, 1000 Ljubljana, Slovenia. [4]Universitá Ca' Foscari Venezia, Via Torino 155, 30170 Venezia, Mestre, Italy. [5]Instituto de Ciencia Molecular (ICMol), Universitat de Valencia, c/Catedrático José Beltrán, 2, Paterna 46980, Spain. ✉e-mail: mattia.benini@tu-dortmund.de; valentin.dediu@cnr.it; mirko.cinchetti@tu-dortmund.de

facial hybridization introduces a spatially random but correlated anisotropy field, leading to a suppression of the conventional domain structure and establishing a frozen disordered ferromagnetic configuration. Notably, this correlated magnetic state is not confined to the surface layer but extends into the ferromagnetic bulk[18].

Despite these emerging insights, most efforts have thus far focused on tailoring the magnetic configuration via proximity effects, while their active control remains largely unexplored. In particular, recent proposals[19] suggest that molecular components could be exploited as active transducers, converting optical excitations into dynamically tuneable proximity effects—opening a pathway towards real-time control of magnetism through molecular excitations.

In this work, we demonstrate such optical functionality by studying Co films in proximity to $C_{60}$ molecules. By selectively exciting excitons in $C_{60}$ with resonant ultrashort light pulses, we dynamically modulate the interfacial hybridization at the Co/$C_{60}$ interface, achieving up to a 60% quenching of the frequency of the dipolar ferromagnetic resonance (FMR) mode. This shift directly reflects modifications in the effective anisotropy field and establishes an effective material platform for optically tuning spin dynamics at GHz frequencies in nanoscale hybrid systems. Given the key role of anisotropy in data storage[20], field-free magnonics[21], and neuromorphic computing[22], our results highlight the potential of molecular optical functionality for next-generation spin-based information technologies.

## Results

### Proximity-induced enhancement of the anisotropy field in Co/$C_{60}$ observed by tr-MOKE

In our investigation, we delve into the optically-induced spin dynamics of Co/$C_{60}$ and reference Co/Al bilayers, examining their dynamical behavior across varying temperatures and applied magnetic fields. The

bilayers consist of thin cobalt films with 5 nm nominal thickness deposited on an $Al_2O_3$ (0001) substrate and covered respectively with 25 nm $C_{60}$ and 3 nm Al. The latter was chosen as a capping layer for its effectiveness in preventing oxidation of the Co layer[23]. Both samples exhibit in-plane magnetic anisotropy, with the out-of-plane direction corresponding to the magnetic hard axis (see Supporting Information – SI – for details).

The experimental setup, as depicted in Fig. 1a, employs a variable out-of-plane magnetic field (**H**) to align the magnetization (**M**) along an effective field (**H**eff), which constitutes the vector sum of the external field (**H**) and an internal field (**H**int), and is accordingly canted out of the sample plane. This internal field is itself the sum of the shape anisotropy field (**H**sh) and the intrinsic sample-dependent anisotropy field (**H**anis). In the tr-MOKE experiments, an ultrafast laser pump pulse transiently perturbs the sample, leading to ultrafast demagnetization within hundreds of femtoseconds and causing **H**eff to deviate from its equilibrium position[24]. Following this initial perturbation, the magnetization experiences damped precession around **H**eff, concurrently undergoing gradual remagnetization (for more details see SI). The observed magnetization precession corresponds to the dipolar ferromagnetic resonance (FMR) mode, characterized by collective oscillations of ferromagnetic (FM) spins with wave vector $k \approx 0$. Within this very well-established experimental framework[23,25,26], the FMR frequency ($\nu$) serves as a sensitive probe of |**H**eff| and, by extension, |**H**anis|, a technologically relevant parameter governing magnetic stability and dynamic response.

For a quantitative analysis, we extracted the oscillation frequency ($\nu$) by subtracting the non-oscillating thermal recovery background and then performing a fit with a damped sine function: $A \sin[2\pi\nu t + \phi_0] \exp(-t/\tau_D)$, where $\tau_D$ represents the damping time. The frequencies extracted using this fit procedure are depicted in

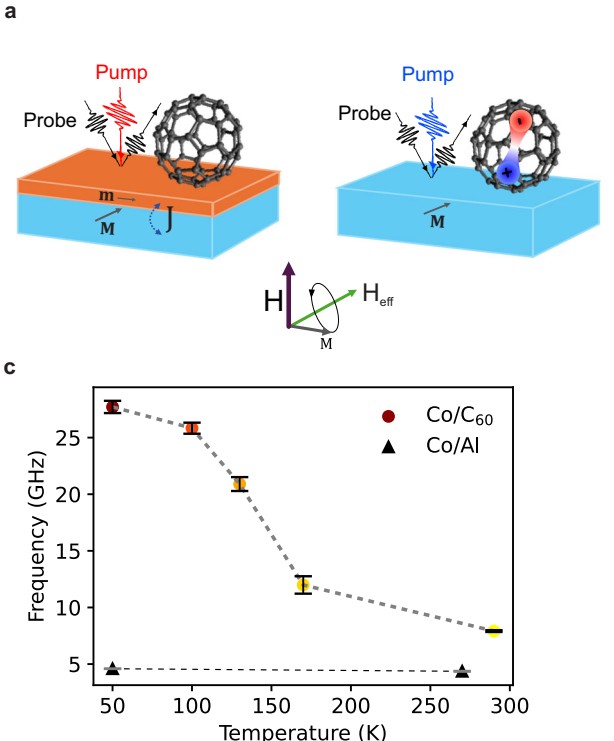

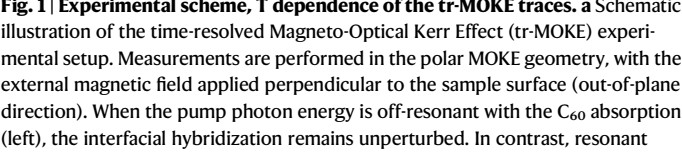

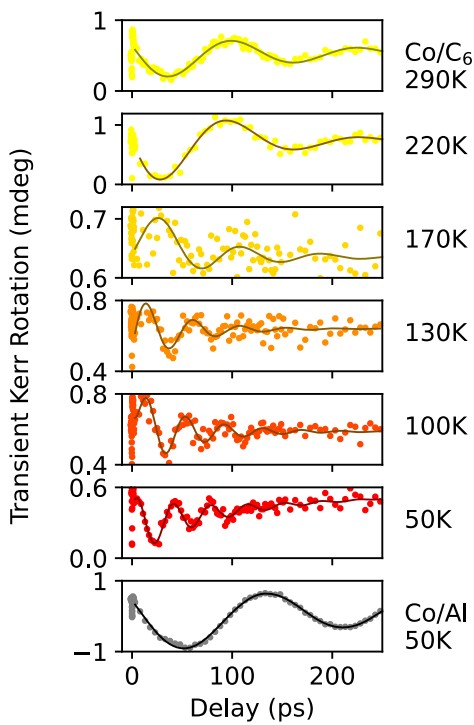

**Fig. 1 | Experimental scheme, T dependence of the tr-MOKE traces. a** Schematic illustration of the time-resolved Magneto-Optical Kerr Effect (tr-MOKE) experimental setup. Measurements are performed in the polar MOKE geometry, with the external magnetic field applied perpendicular to the sample surface (out-of-plane direction). When the pump photon energy is off-resonant with the $C_{60}$ absorption (left), the interfacial hybridization remains unperturbed. In contrast, resonant

excitation of excitons in $C_{60}$ (right) modifies the hybrid interface, altering the magnetic response. **b** tr-MOKE signals measured for $H_{ext}$ = 0.5 T as a function of the temperature for the reference Co/Al sample and the Co/$C_{60}$ sample, respectively. **c** Values of the precession frequency ($\nu$) extracted from the data in **b** as described in the main text. Error bars are obtained by fitting procedure.

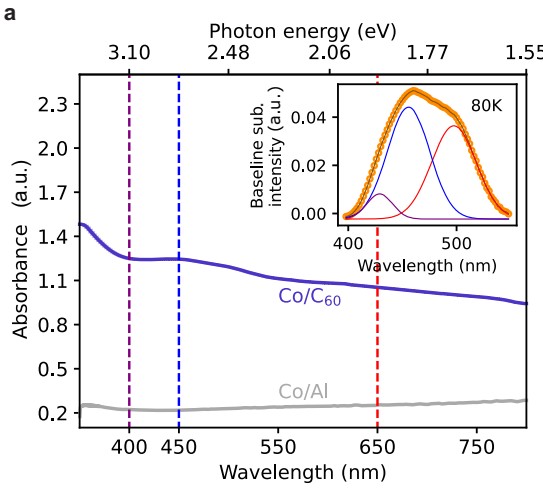

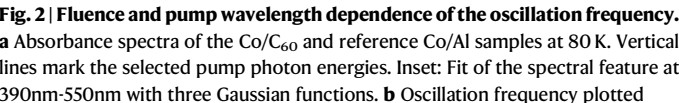

**Fig. 2 | Fluence and pump wavelength dependence of the oscillation frequency.** **a** Absorbance spectra of the Co/C$_{60}$ and reference Co/Al samples at 80 K. Vertical lines mark the selected pump photon energies. Inset: Fit of the spectral feature at 390nm-550nm with three Gaussian functions. **b** Oscillation frequency plotted against the absorbed energy density in the Co layer for all selected pump wavelengths, with linear fits depicted by the lines. Error bars are obtained as indicated in the SI sect. I.

Fig. 1b as a function of the temperature. As the temperature decreases, the frequency difference between Co/C$_{60}$ and Co/Al increases markedly. At low temperatures, the Co/C$_{60}$ system shows an enhancement in $\nu$ by nearly a factor of five, indicating a substantial increase in the anisotropy field induced by the proximity of C$_{60}$. This observation is consistent with previous reports of enhanced $H_{anis}$ values[18,27], attributed to hybridization between the Co electronic states and the C$_{60}$ molecular orbitals[11,13], which substantially modifies the interfacial magnetic anisotropy.

To place these findings in a broader context, we note that the observed proximity-induced modulation of spin dynamics is rooted in a universal interfacial mechanism recently identified across multiple cobalt/molecule heterostructures[17,28]. A systematic tr-MOKE investigation revealed the emergence of a strongly anisotropic interfacial magnetic layer, which is driven by chemical hybridization between Co surface $3d$-orbitals and molecular π-systems. The resulting interfacial layer, with magnetization **m** (Fig. 1a), has been recently shown to be fundamentally different from the magnetic structure of the underlying magnetic layer with magnetization **M** and a conventional uniaxial magnetic anisotropy. The formation of this interface dramatically modifies the magnetic state of the whole ferromagnet, leading to the emergence of a Correlated Ferromagnetic Glass[17], with a correlated random anisotropy term $K_R$ induced by surface hybridization with the molecular orbitals.

To describe the dynamics of such state in the ps timescale, the following free energy functional was recently proposed[28]:

$$F = \mu_0 \xi_C^2 \sum_\beta \frac{(\nabla M_\beta)^2}{2} - \mu_0 \mathbf{H} \cdot \mathbf{M} + \frac{K_\perp}{2} (\mathbf{M} \cdot \mathbf{e}_\perp)^2 - K_R \left| \frac{\mathbf{M}}{M_0} \cdot \mathbf{e}_R \right|^\alpha \quad (1)$$

The first three terms of the right-hand side correspond to exchange, Zeeman, and out-of-plane anisotropy energies. The last term is a phenomenological expression capturing the effect of the surface hybridization on the cobalt layers. It contains the anisotropy energy density parameter $K_R$ whose easy local direction $\mathbf{e}_R$ is random but correlated over a lengthscale defined by a correlation radius $r_C$. This term governs the anisotropy field around which the magnetization **M**, coupled to **m** via exchange interaction (J in Fig. 1a), precesses in our experiments. Due to the coupling, the interface layer follows the precession of **M** adiabatically[28]. As a result, the precession frequencies extracted in our experiments are directly sensitive on the parameters

$K_\perp$, $K_R$ and $\mathbf{e}_R$, and thus provide a means to monitor the interfacial modifications induced by C$_{60}$ hybridization.

Building on this framework, our work demonstrates for the first time that interfacial hybridization−as captured by $K_\perp$, $K_R$ and $\mathbf{e}_R$ −can be dynamically controlled via resonant optical excitation. We show that it is possible to optically quench the Co−C$_{60}$ hybridization by resonant exciton formation in C$_{60}$, and as a result to modulate the correlated random anisotropy field on picosecond timescales. As schematically illustrated in Fig. 1a, this capability introduces an additional degree of control over proximity-induced magnetic phenomena, with promising implications for the development of ultrafast spintronic devices.

## Pump wavelength dependence of the oscillation frequency

Our main goal is to actively manipulate the interface-driven modifications in magnetic anisotropy by optically exciting the Co/C$_{60}$ hybrid units, utilizing the same pump pulse responsible for initiating the ultrafast magnetization dynamics within the Co layer. Optical modification of the magnetic anisotropy will then lead to a modification of the FMR precession frequency that we detect in our tr-MOKE experiments. To accomplish this, we conducted an experimental sequence where we captured several tr-MOKE signals at the same out-of-plane magnetic field utilized in Fig. 1c ($\mu_0 H = 0.5$ T), varying the pump fluence and the photon energy of the pump laser. From these signals, we extracted the oscillation frequency and the precession decay time for each set of conditions, facilitating a comparative analysis of the results. To ascertain that the variations in the magnetization dynamics were attributable to the excitation of the molecular overlayer and not solely to the Co layer, we replicated the experiments on the reference Co/Al sample.

To select the most effective excitation wavelengths, we first conducted temperature-dependent absorbance measurements on the Co/C$_{60}$ system in the wavelength range 350-800 nm (3.5 eV−1.5 eV). Additionally, we analysed the Co/Al system to identify any absorption peaks characteristic of Co. We anticipated that most of the Al would be oxidized after exposure to ambient conditions[23], thus contributing minimally to the measurements. The absorbance spectra measured at 80 K are shown in Fig. 2a, with the full dataset available in the supplementary material (SI). Compared to Co/Al, the absorbance spectrum of Co/C$_{60}$ shows a distinct peak emerging below 400 nm. This feature corresponds to a peak at 355 nm, resulting from the first allowed optical transition of isolated C$_{60}$ molecules[29–34]. The

intramolecular $S_0 \to S_1$ absorption, being symmetry forbidden, appears as a very weak feature in the spectrum at 650 nm. In the spectral range between 390 nm and 550 nm, the spectrum is dominated by a broad feature attributed to intermolecular excitations, which include both charge transfer and localized Frenkel excitations. For longer wavelengths, only localized excitations contribute to the spectrum[32]. The inset of Fig. 2a shows the fit of the broad feature at 390 nm-550 nm with three Gaussian functions. The SI includes the temperature dependence of the extracted peak position and FWHM. We observe a non-monotonic behavior, with a change in slope occurring between 80 K and 120 K. This behavior is explained by a structural transition in the $C_{60}$ layer around this temperature[35], which in turn influences the properties of the charge transfer excitons in this spectral range.

Given these findings, we have chosen two wavelengths in the spectral range of the broad absorption feature related to intermolecular excitations for resonantly exciting the Co/$C_{60}$ system and tuned the pump wavelength to these values: 450 nm (2.75 eV) and 400 nm (3.1 eV). In addition, we have chosen 650 nm (1.9 eV) to assess the response of the system in the presence of a very weak intramolecular excitation. For the probe beam, we selected the wavelength of 800 nm (1.5 eV), which is in the transparency window of $C_{60}$, ensuring that the observed response is specifically sensitive to the Co layer dynamics. Moreover, systematic studies of Co/molecules heterostructures[28] indicate that the Co/molecules interfacial layer primarily modulates the effective anisotropy field of the Co layer rather than exhibiting independent spin dynamics.

We conducted tr-MOKE experiments on both Co/$C_{60}$ and Co/Al samples using the three selected pump photon energies, analyzing the results as a function of the absorbed energy density in the Co layer ($w_{Co}$). This parameter, which accounts for wavelength-dependent absorption differences, is determined in the SI. Additionally, we used the maximal demagnetization peak as a verification parameter to ensure the effective laser fluence was correctly chosen. The recorded data are presented in the SI. In Fig. 2b, we report the values of $\nu$ extracted from the data in the low pump-fluence region (up to 0.15 mJ/cm$^2$). In this fluence range, the Co/$C_{60}$ precession frequency shows a linear dependence on $w_{Co}$, whereas in the reference Co/Al sample, the frequency remains constant and agrees well with the pristine Co resonance frequencies. (Exemplarily data for the Co/$C_{60}$ system at higher fluences are reported in the SI.)

By performing linear fits, we extracted the extrapolated values of $\nu$ at $w_{Co} = 0$ ($\nu_0$) and the slopes $s = \frac{d\nu}{dw_{Co}}$. For $\nu_0$ we obtained virtually identical values (within the error bars) for the three pump wavelengths: (26.0 ± 0.8) GHz, (24.6 ± 0.8) GHz, and (25.8 ± 0.8) GHz for 650 nm, 450 nm, and 400 nm, respectively. We therefore interpret $\nu_0$ as the FMR frequency of the pristine Co/$C_{60}$ system, i.e. for the case where the magnetization is tilted adiabatically out of its equilibrium position, without altering the electronic properties of the system itself. Moving to the extracted slopes ($s$), we observe a significant dependence on the pump wavelength: (−22 ± 8) GHz/(mJ/cm$^2$), (−158 ± 27) GHz/(mJ/cm$^2$), and (−91 ± 16) GHz/(mJ/cm$^2$) for 650 nm, 450 nm, and 400 nm respectively, with a much larger slope observed for pump photon energy in the intermolecular excitations peak. Importantly, in the Co/Al sample, no difference emerges between the two tested pump wavelengths, indicating that the pronounced wavelength dependence of $s$ in the Co/$C_{60}$ sample is attributable to the excited state of the molecular layer.

## Discussion
### Exciton influence on the spin dynamics
We now move to the discussion of the experimental results. As already mentioned, we interpret the $\nu_0$ value as the FMR frequency of the electronically unperturbed Co/$C_{60}$ layer. Regarding the observed dependence of $\nu$ from $w_{Co}$, we observe that when the pump photon

energy is in the region of the intermolecular absorption (resonant pump) the decrease of $\nu$ is much steeper than for the pump at 650 nm (off-resonant pump). In general, increasing $w_{Co}$ leads to a progressing decrease of the precession frequency of Co/$C_{60}$ toward the precession frequency of the pristine Co sample. If the pump is in resonance, much lower values of $\nu$ can be achieved than for the off-resonant case. Since the precession frequency is proportional to the anisotropy field, the observed behavior indicates that resonant pumping effectively reduces the anisotropy field of the Co/$C_{60}$ system towards the value of the pristine Co sample. The higher the pump fluence, the stronger the quenching effect. Therefore, we conclude that the effect of resonant pump photons is to quench the hybridization at the Co/$C_{60}$ interface, causing it to behave magnetically similar to the pristine Co layer. This quenching effect underscores the impact of resonant molecular excitations on the magnetic properties of the hybrid FM/Molecule system, providing insights into the tunability of magnetic anisotropy through optical excitation. To quantify the effect of this exciton-mediated optical control of GHz spin dynamics, we quantify the modification of the precession frequency at $w_{Co} = 0.07$ mJ/cm$^2$. At this energy density, we note a significant modulation of more than 60% between the resonant and off-resonant pumping. This highlights the profound impact of resonant optical excitation on the spin dynamics and magnetic anisotropy in molecule-interfaced Co thin films.

Before concluding, we turn to the mechanism leading to the optically induced quenching of hybridization at the Co/$C_{60}$ interface and the involved timescales. To shed light on this point, we performed fs-resolved transient reflectivity experiments following excitation with (351 ± 10) nm pump pulses, recorded between 0 and 12 ps in the wavelength range of 450–700 nm. The measurements, together with the line profiles extracted from those measurements at different time delays of 0.5 ps, 2 ps, 5 ps, are shown in Fig. 3a, b, respectively. In these spectra, we observe a negative peak around 500 nm, followed by a positive peak at 540 nm. This feature has been attributed in the literature to the presence of local electric fields generated by the direct population of intermolecular charge transfer states and their associated strong electric dipoles[32].

Excitation at 351 nm leads to the formation of charge transfer excitons[33,34]. In the Supplementary Information we extract the timescales related to exciton dynamics, that we summarize in Fig. 3c. The charge-transfer excitons decay on a timescale of $\tau_1 = 0.18$ ps to lower-lying charge transfer (CT) states or localized Frenkel excitons. The formed Frenkel excitons, as well as those directly excited by the pump laser, decay with a time constant of $\tau_2 = 4.5$ ps. Finally, the relaxed Frenkel excitons decay with a much longer time constant, in agreement with the 150 ps reported in literature[32]. This latter relaxation time scale is well within the damping time of the oscillations observed in the Co/$C_{60}$ system (see SI for details on the extraction of the time constants from the experimental data as well as on the damping constant of the spin dynamics in Co/$C_{60}$). It is also observed when pumping off-resonance with 650 nm[32]. However, in this case, Frenkel excitons can only form via direct laser excitation, and since intramolecular $S_0 \to S_1$ absorption is symmetry forbidden, significantly fewer Frenkel excitons are formed during off-resonant excitation.

The observed exciton dynamics lead us to the following explanation for the experimental data in Fig. 2b: using resonant pump excitation, we create charge transfer excitons in $C_{60}$. When the pump pulse is absorbed by the $C_{60}$ molecules, charge transfer (CT) excitons are formed within the first few hundred femtoseconds, rapidly decaying (in a few picoseconds) into Frenkel-like excitons ($S_1$) that remain stable for several tens of picoseconds, the same timescale on which we observe the coherent spin dynamics in Co/$C_{60}$. These excitons consist of a hole in the Highest Occupied Molecular Orbital (HOMO) of the molecule and an electron in the Lowest Unoccupied Molecular Orbitals (LUMO), specifically the LUMO + 1 for CT excitons or LUMO for $S_1$ excitons[36].

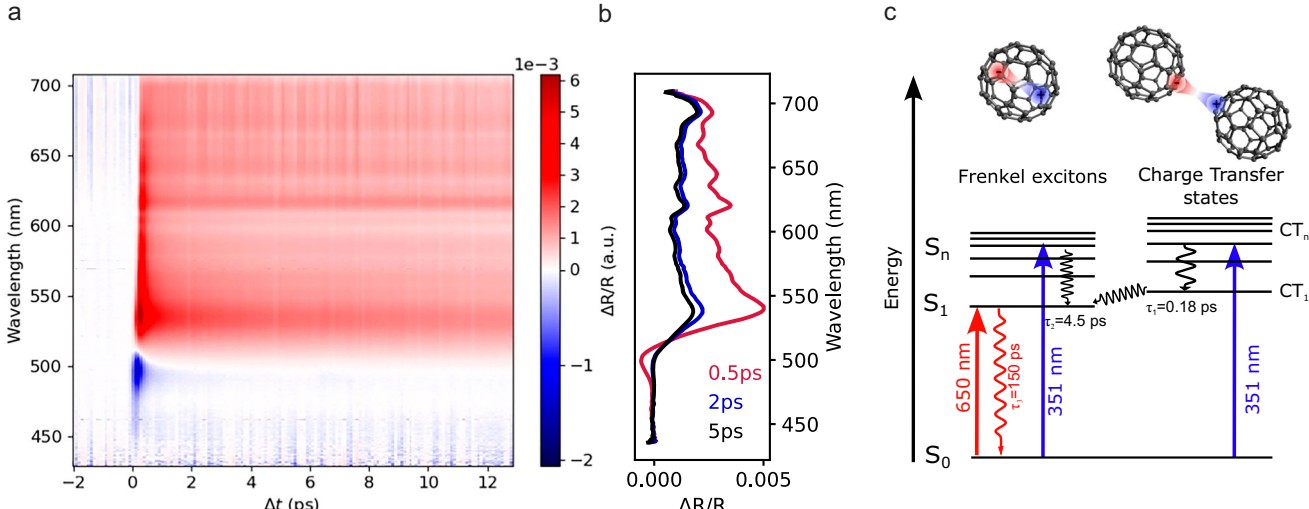

**Fig. 3 | transient reflectivity and exciton energy diagram. a** Femtosecond-resolved transient reflectivity spectra of Co/$C_{60}$ following on-resonance excitation with 351 nm pump pulses ($\Delta t$ is the pump-probe time delay). **b** Line profiles extracted from the data in **a**. **c** Exciton scheme of $C_{60}$.

Although the complex magnetic structure of the Co/$C_{60}$ interface has only recently been elucidated[17], on a local scale, the chemisorption of $C_{60}$ molecules on the Co surface can be understood through the formation of hybrid molecule-metal $d_z^2$ bonds[13,14], which strongly affect the surface properties[12,14]. Crucially, only the few C atoms in direct contact with the surface contribute to such hybridization, as indicated by the bonds depicted in Fig. 4a. In contrast, the other C atoms interact weakly with the surface owing to the "soccer ball" molecular structure. This is seen in the density of states in Fig. 4b (see SI for more information). Consequently, $C_{60}$ exhibits dual properties: it retains its molecular character, allowing it to display excitonic features, while, through the few hybridized C atoms, it can significantly alter the surface's magnetic properties. When the Co/$C_{60}$ sample is pumped with resonant photons, the long-living Frenkel-like excitons formed in $C_{60}$ effectively result in an excited molecular state which in turn modifies the molecular properties and, therefore, the properties of the molecule-surface interaction. Specifically, our observations indicate that this modification partially quenches the interfacial hybridization strength of the entire Co/$C_{60}$ system. The higher the absorbed fluence, the stronger the quenching effect becomes, causing the Co/$C_{60}$ hybrid units to behave more like pristine cobalt. Consequently, as the quenching effect intensifies, the anisotropy field of the Co/$C_{60}$ system approaches that of pristine Co films (which is lower). This explains the observed reduction in oscillation frequency with increasing absorbed fluence.

Given that the exciton lifetime is approximately 150 ps, comparable to the timescale during which $k$ - 0 spin waves can be detected, the exciton density undergoes an exponential decay, which suggests a potential time dependence of the oscillation frequency. To investigate this effect, we conducted an extended analysis of our experimental data, revealing that the oscillation frequency is indeed time-dependent. The frequency values reported in the main manuscript correspond to the early-time oscillations, where the exciton density is still high and before significant decay occurs. For times approaching the exciton lifetime, deviations from a purely sinusoidal behavior emerge, indicating a gradual frequency shift (see Fig. S5 in the SI). This time-dependent frequency evolution does not contradict our primary claim—that the FMR frequency can be optically tuned by selectively exciting the $C_{60}$ layer. Rather, it reinforces the transient nature of the exciton-driven modulation, further supporting the role of molecular excitations in controlling interfacial magnetic properties. Our results demonstrate that not only the modulation is strong but also localized at the nanoscale, as it originates from complex quantum behaviors.

Moreover, we propose that the underlying physics is universal, potentially allowing for the optical tuning of any proximity-induced physical property, beyond just magnetic ones. To further substantiate the optical tunability of proximity-enhanced functionalities in hybrid systems, future experiments could explore the role of interfacial hybridization more systematically by introducing spacer layers with varying electronic coupling strength between $C_{60}$ and the ferromagnetic substrate. These control measurements would allow disentangling proximity-induced effects from other possible contributions. While such studies go beyond the scope of the present work, they represent a compelling direction for validating and generalizing the proposed mechanism across a broader class of molecular interfaces.

## Methods

### Sample fabrication

Thin Co (5 nm thickness) films were deposited by electron beam evaporation on $Al_2O_3(0001)$ substrates at room temperature and base pressure of $1.1 \times 10^{-10}$ mbar. The organic layer or the Al layer were subsequently deposited on top of the Co layer by thermal evaporation (base pressure $1.1 \times 10^{-9}$ mbar) at room temperature without breaking the vacuum.

### Setup for absorbance measurements

The static absorbance measurements were performed using a commercial spectrophotometer (CARY 6000i, Agilent Technologies) in transmission. The sample was mounted in a He-flow cryostat (Oxford Instruments) for temperature-dependent measurements with a precision of ±1 K. For the baseline correction, an identical sample holder was placed in the reference beam.

### Setup for time-resolved magneto-optical Kerr effect measurements

The tr-MOKE measurements were done by means of two different table-top setups for optical pump–probe measurements. The fixed-fluence characterization was performed with a two-color pump-probe setup based on a high repetition-rate (250 kHz) 50-femtosecond Ti:Sapphire laser amplifier and a split-coil superconducting 7 T optical magnet with variable temperature He exchange gas sample insert. A part of the output pulse train was frequency doubled (λ = 400 nm, 3.1 eV photon energy) to derive the pump pulses while the probe pulses were derived from the remaining fundamental pulse train (λ = 800 nm, 1.55 eV). The reflected-probe-beam transient polarization

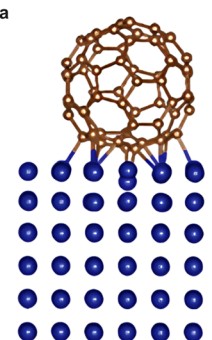
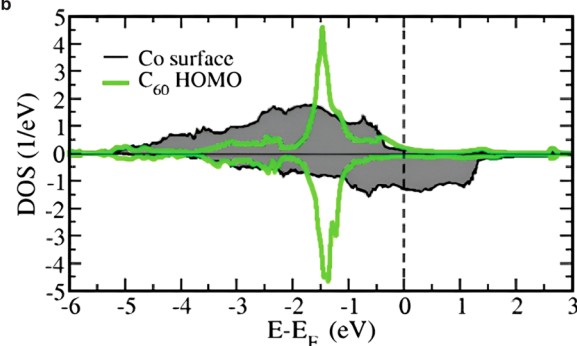

**Fig. 4 | DFT calculations. a** $C_{60}$ adsorbed on the Co surface. **b** Co and $C_{60}$ HOMO-PDOS, calculated by DFT. Positive (negative values) are for spin-up (spin-down) states. The Co surface DOS is obtained by summing the data for all eighteen surface cobalt atoms in the supercell. Both the HOMO- and CO-PDOS are normalized so that their integral over the energy is equal to one.

rotation was detected by means of a balanced detection using a Wollaston prism and a pair of silicon PIN photo diodes. The pump beam was modulated with an optical chopper at ~2.5 kHz and a standard lock-in detection scheme was used to acquire the photodiodes differential signal.

The tr-MOKE measurements with variable pump wavelengths were done by means of another setup with the pump and probe light pulses independently tuneable in the range 0.5–3.5 eV. The setup described in refs. [37,38]. The probe beam was kept fixed at $\lambda_{PR} = 800$ nm with 0.2 mW fluence, with a 30 μm spot diameter and repetition rate of 100 kHz. The pump spot diameters were: 36 μm ($\lambda_{PU} = 650$ nm), 50 μm ($\lambda_{PU} = 450$ nm) and 50 μm for ($\lambda_{PU} = 400$ nm); and the repetition rate was 50 kHz. The duration of both pump and probe pulses was ≈ 50 fs calculated as the FWHM (corresponding to a spectral energy spread, ΔE = 0.04 eV).

### Setup for ultrafast transient reflection spectroscopy
A femtosecond laser beam was focused on a 1 mm sapphire crystal to generate a pulsed white-light probe with a repetition rate of 100 kHz. The probe beam was reflected from the sample and dispersed by a diffraction grating before being focused on a 1D CMOS array (Stresing GmbH). The pump beam, synchronized with the probe beam, was modulated with an optical chopper, which was triggered at one-256th of the laser repetition. In the pump-probe scheme, the probe beam was detected at a frequency of one-128th of the laser repetition, allowing the isolation of the signal at the photoexcited state. With this technique, we acquire the differential reflectivity ΔR/R data as a function of probe wavelength and time delay.

### Computational details
The calculations are performed by using an implementation of DFT based on the Green's function technique. Specifically, we employ the SMEAGOL code[39] that obtains the Kohn-Sham Hamiltonian from the SIESTA package[40]. We assume Co to have fcc structure. The $C_{60}$ molecule is adsorbed on Co in the so-called "hexagon-pentagon" configuration, as described in ref. [13]. The surface region is contained in a $4 \times 4$ supercell, comprising the $C_{60}$ molecule and six Co atomic layers, which are coupled to a semi-infinite Co bulk region through an embedding self-energy[41]. Periodic boundary conditions are assumed along the in-plane directions. Core electrons are treated with norm-conserving Troullier-Martins pseudopotentials[42]. The valence states are expanded through a numerical atomic orbital basis set including multiple-z and polarized functions. The local spin density approximation (LSDA)[43] is assumed for the exchange correlation functional. The electronic temperature is 300 K. The real space mesh is set by an equivalent energy cutoff of 300 Ry. We

use a $6 \times 6$ k-point mesh in the two-dimensional surface Brillouin zone. The DOS projected on the HOMO shown in Fig. 4b is obtained using the algorithm in ref. [44].

### Data availability
The data used in this study are publicly available in the Zenodo database with the following link: https://doi.org/10.5281/zenodo.15720805.

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

## Acknowledgements

We acknowledge the help of Cristian Manzoni for the improvement of the ultrafast transient reflectivity setup. This work was supported by the EC H2020 program under grant agreement No. 965046, FET-Open project INTERFAST (Gated interfaces for fast information processing) and by the European Research Council (ERC) under the European union's Horizon 2020 research and innovation programme (Grant agreement No. 725767-hyControl). We also acknowledge support of the Deutsche Forschungsgemeinschaft (DFG) through the project Proximity, Project number: 556408835. A. Shumilin acknowledges the financial support from the European Union (ERC-2021-StG-101042680 2D-SMARTiES). T. Mertelj and V. A. Dediu acknowledge support within the Programme on Scientific Cooperation between the National Research Council of Italy and the Jožef Stefan Institute.

## Author contributions

M.B. T.M., V.A.D. and M.C. mostly contributed to the drafting of the manuscript. M.C. and V.A.D. contributed to the coordination of all experimental work, R.K.R. and M.B. performed all sample preparation, M.B., J.S. U.P., S.B., and F.M. performed tr-MOKE characterizations, M.B. and J.S. performed data analysis of the tr-MOKE traces, E.Z. and U.P. performed absorbance measurements and relative data analysis, U.P., R.L. and D.G. performed the transient reflectivity measurements and analysed the relative data, A.D. performed DFT modeling, A.S. developed the micromagnetic model, I.B., A.S. and all other authors contributed to the interpretation of the experimental results.

## Funding

## Competing interests

The authors declare no competing interests.
