## [Transparent Peer Review file · Nature Communications]

Light-driven modulation of proximity-enhanced functionalities in hybrid nano-scale systems

Corresponding Author: Professor Mirko Cinchetti

Version 0:

Reviewer comments:

Reviewer #1

(Remarks to the Author)

The authors report on employing optical pulses to transiently modulate the magnetic anisotropy of a hybrid cobalt/C60 bilayer. They attribute the modulation to exciton excitation in the C60 layer, which supposedly reduces orbital hybridization, thereby altering the anisotropy of the cobalt layer. Time-resolved magneto-optical Kerr effect (tr-MOKE) measurements are employed to monitor this modulation by analyzing the precession frequency of the magnetization after optical pumping. The authors observe the strongest modulation of magnetic anisotropy when the pump wavelength is resonant with the C60 excitons, emphasizing the role of optically excited excitons. The comparison between Co/C60 and a control Co/Al sample, which lacks these trends, further suggests the involvement of excitons in the observed phenomena.

My assessment of this work is mixed. It presents some interesting findings, particularly the modulation of gigahertz magnetic oscillation frequency via optical pumping. I agree with the authors on the central role of excitons in this phenomenon as the comparison between the Co/C60 and Co/Al samples supports the central hypothesis. However, the details of the modulation mechanism are not sufficiently elaborated, leaving several important unanswered questions which require additional arguments, experiments, and controls to fully substantiate the observations. First, the nature of the oscillations is not clearly identified. The authors frequently describe them as “oscillations” or “precession” without specifying their exact origin. Are these spin waves or simply ferromagnetic resonance (FMR) modes of the samples? Clarifying this distinction is crucial for a comprehensive understanding of the results (see detailed comments below). Second, the spatial location of these oscillations remains unclear. Are they confined to the cobalt layer, the interface, or perhaps within the C60 layer itself? The manuscript seems to implicitly assume that the cobalt layer hosts these oscillations. While this is a reasonable hypothesis, it is not definitively proven and requires confirmation through additional experiments (see detailed comments below). Third, the technological significance of the study appears overstated. For instance, the authors claim, “This approach achieves up to 60% modification in precession frequency, which can be related to a modification of the anisotropy field, showcasing a groundbreaking method for optical control of a technologically relevant parameter in nanoscale hybrid units.” This assertion is exaggerated for several reasons. The reported modulation is only achieved at low temperatures, which are rarely of practical interest for technological applications, and requires high laser fluences. Moreover, the claimed 60% modulation comes at the expense of a significantly shortened decay time. Supporting information (Figure S4) shows that at a laser fluence of 0.1 mJ/cm², where the 60% modulation is observed, the decay time is nearly halved. Additionally, modulation schemes based on optical pulses are inherently challenging to integrate into practical technologies. Consequently, the claims regarding the “groundbreaking” and “technological” importance of this study do not appear to be well-supported.

Overall, there are some interesting elements in this study, but not well substantiated. Thus, I suggest giving authors an opportunity to revise the manuscript for a second round of revision. In any case, I recommend considering the following points:

1. As demonstrated in [Physical Review B 90, 125311–6 (2014)], the proximity of C60 to cobalt can lead to the magnetization of the C60 layer through spin-polarized charge transfer from Co. Consequently, shortly after optical pumping, the C60 layer itself may become magnetized. This raises the possibility that the observed oscillations originate directly from the C60 layer rather than solely from the cobalt layer. Given the transient nature of the C60 magnetization, these oscillations would also be transient, which could explain the observed decaying behavior. The authors should explore this possibility in

detail, supported by appropriate experiments or simulations, to provide a clearer description of the origin of the oscillations. A systematic variation in the thickness of the Co and/or C60 layers could offer valuable insights into the source of the oscillations and help clarify whether they arise from the cobalt layer, the C60 layer, or an interfacial effect.

2. Throughout the manuscript, the authors use general terms such as “oscillation,” “spin dynamics,” and “precession” without specifying the precise nature of the observed oscillations. Are these oscillations attributable to ferromagnetic resonance (FMR) modes of the cobalt layer, or do they correspond to spin waves? Clarifying this distinction is critical. For instance, could the observed differences between the Co/Al and Co/C60 samples be explained by a scenario where the Co/Al sample primarily exhibits FMR modes, while the Co/C60 sample supports high-frequency spin waves? This hypothesis might also account for the higher damping observed in the Co/C60 sample compared to Co/Al. The granular and semiperiodic structure of the C60 “soccer balls” at the top layer could act as a grating, providing the necessary k-matching condition to excite high-frequency spin waves at non-zero wave vectors. In contrast, this mechanism would be absent in the Co/Al sample, which might explain why only low-frequency FMR modes are accessible in that case. The authors should address these points and consider additional experiments or simulations to confirm the nature of the oscillations.

3. As demonstrated in [Phys. Rev. Lett. 88, 227201 (2002)], the optical excitation of magnetic oscillations typically occurs in multiple stages. Initially, light absorption leads to rapid heating of the magnetic material, which causes both demagnetization and changes in magnetic anisotropy, particularly shape-induced anisotropy. According to the referenced study (see Figure 1b), coherent magnetic oscillations only become observable after the heat is dissipated into the substrate, a process that takes several picoseconds, often up to ~10 ps. In contrast, the present manuscript reports the emergence of magnetic oscillations immediately following optical excitation, at pump-probe delays of approximately 0 picoseconds. Could the authors explain this discrepancy? It appears that the oscillations in the current study occur in a system that is thermally out of equilibrium. If this is indeed the case, one might expect the oscillation frequency to exhibit time dependence as the system cools down gradually. However, the reported oscillations seem to be time-independent. This apparent contradiction should be addressed and clarified in the manuscript.

4. The main claim is that excitons are modulating the magnetic anisotropy, and the change of anisotropy changes the oscillation frequency. Given the exponential decay of excitons density (after optical excitation), shouldn't we expect an exponential change of the oscillation frequency during the exciton lifetime?

5. Page 2; Authors states: “...we investigate the potential for data storage, magnetic sensing, and quantum computing applications.” I don't see any of these applications being investigated in this study. Can authors further elaborate what part of the manuscript this sentence is referring to?

6. Page 2; Authors write: “While ultrafast optical control has been demonstrated in dielectric materials⁷, metallic systems remain challenging due to rapid electron-hole pair dephasing and coherence loss caused by Coulomb screening⁸.” This statement does not sound accurate to me. A careful literature survey shows multiple examples of successful ultrafast and coherent optical control in metallic systems, see for instance [Science 382, 299–305 (2023)]. Authors need to conduct a better literature survey.

7. The authors have primarily focused on the oscillation frequencies under various conditions (e.g., temperature, pump fluence) while largely neglecting the trends observed in the decay rates and, more critically, the amplitudes of the oscillations. A particularly concerning issue is the apparent lack of correlation between pump fluence and oscillation amplitude. For instance, as shown in Figure S3 for a 650 nm wavelength, increasing the pump fluence by a factor of four (from 0.016 to 0.071) results in only a minimal change in the oscillation amplitude. Similar discrepancies are observed in other measurements. One would typically expect a linear increase in amplitude with pump fluence, at least in the low-fluence regime, due to the direct relationship between absorbed energy and excitation strength. These observations raise questions about the underlying mechanisms and suggest that key aspects of the data have not been fully analyzed or discussed. The authors should carefully process and examine these trends in the revised manuscript, providing a detailed explanation of the observed behaviors and their implications.

8. Following the above point; There is a very abstract but not conclusive discussion in the supporting information on decay rate with pump fluence. The trend for Co/C60 samples is systematic at all tested wavelengths, showing a significant increase in the decay rate. For instance, at 650nm, the decay time drops from ~ 50ps to ~ 12ps as the pump fluence reaches 0.12mJ/cm². Interestingly a completely opposite trend is observed for the Co/Al sample. This seems too systematic to be ignored in the analysis of the modulation mechanism.

9. Page 4; referring to Figure 1, Authors express “...This set of measurements demonstrates that we can use the frequency of the GHz oscillations extracted from the tr-MOKE experiments as a measure of the anisotropy field ...”. For a more quantitative calculation, authors can also leverage the field dependence of the oscillations and use the Standard Kittel formula for FMR frequency and back calculate the exact anisotropy field/energy of Co/C60. Is there any reason this has not been done here?

10. Page 9; Authors write “... we observe the modification of the precession frequency at $w_{Co} = 0.1$ mJ/cm². At this energy density, we note an extremely significant modulation of more than 60% between the resonant and off-resonant pumping” . However, it's not clear how the 60% modulation is calculated. The resonant pumping is 450nm wavelength condition. Looking at Figure 2b, no data point is provided at 0.1 mJ/cm² for this wavelength.

11. Figure 2a; Authors should include the absorption spectra of the pure C60 layer with the same thickness as the one used on Co/C60 experiments? This will enable a better understanding of the optical response of the C60 independent of the interfacial effects from the Co layer.

12. Figure 2a; can authors comment on the step-like jump in the optical absorption obtained from Co/Al in the wavelength range between 550-600nm?

13. Figure 1; Figures 1c and 1d show that the oscillation frequency of the Co/C60 sample is significantly higher than that of the Co/Al sample, differing by a factor of approximately 5–6 at low temperatures. The authors attribute this difference to the stronger anisotropy in the Co/C60 sample compared to Co/Al. However, this substantial difference in anisotropy is not reflected in the magnetization curves presented in Figure 1b, where both samples reach their saturation values at nearly the same magnetic field. This observation is counterintuitive, as the saturation field is directly influenced by the anisotropy field, which should differ significantly between the two samples if the stated explanation holds.

14. The central focus of the manuscript is on magnetic anisotropy and its modulation for technological applications. However, none of the cited works adequately highlight the significance of magnetic anisotropy in spintronic applications. Including references to studies where anisotropy serves as a critical enabler for Spintronics could strengthen the manuscript. Here are some examples of the critical role of magnetic anisotropy in Data storage [Adv. Mater. Technol. 2023,8,2300676], Field-free magnonics [arXiv:2411.14428, <https://doi.org/10.48550/arXiv.2411.14428>], and neuromorphic computing [Phys. Rev. Applied 19, 064018 – Published 6 June, 2023].

Reviewer #2

(Remarks to the Author)
see attached file

Reviewer #3

(Remarks to the Author)

Benini et al. reports on a novel light-driven modulation carried out to introduce and control proximity-enhanced functionalities in a hybrid Co/C60 system.

The main scientific idea of the work is innovative and definitely brings a lot of new insight in the understanding and manipulation of the above-mentioned proximity effects. In fact, even if some of the most established properties of proximity effects have been reported and discussed, their full potential still needs to be unleashed by controlling and not just creating functionalities. Cinchetti and their team make a substantial advance in this regard by demonstrating the optical functionality of Co films in close proximity to C60 macromolecules. By generating excitons in C60 by means of refined resonant ultrashort light pulses, Benini et al. managed to significantly tune the hybridization, therefore leading to a novel approach to modulate GHz spin dynamics in the Co layer.

The work very much supports their claims through a set of very carefully thought and carried out experiments backed up by a solid theoretical/computational section.

However, the paper needs some revisions:

- The abstract needs to be substantially beefed up to present better the findings of the authors' study. As it is, its content is vague.
- The conclusions paragraph in the final part of the article is rather dry and does not provide the reader with any perspective on future usage of the authors' findings. I encourage the authors to rework this final part and emphasize the technologically relevant aspects of their work.

Version 1:

Reviewer comments:

Reviewer #1

(Remarks to the Author)
Please see the attached PDF.

Reviewer #2

(Remarks to the Author)

I have reviewed the revised manuscript. The authors have satisfactorily resolved all the concerns raised. Consequently, I recommend its publication in NC.

Reviewer #3

(Remarks to the Author)

The authors have successfully implemented the required changes. The article is now way more accessible to a broader audience of scientists, and its outlooks are remarkably clearer in the revised version.

Version 2:

Reviewer comments:

Reviewer #1

(Remarks to the Author)

have carefully read the revised manuscript and noted noticeable changes throughout. In particular, I appreciate the addition of new material, which reduces the manuscript's reliance on other arXiv papers and makes it more self-sufficient. However, the manuscript still rests on some not-fully-substantiated claims, and critical control experiments—such as inserting an intermediate layer between C60 and cobalt (e.g., a copper spacer)—have not been pursued by the authors. As a result, some of the arguments remain somewhat hand-wavy and fall short of a fully rigorous technical support.

A piece of advice to the authors: I encourage you to avoid using harsh or dismissive language when responding to technical questions and scientific comments (it might simply be lack of proficiency in English). Phrases such as “fundamentally flawed” or references to “standard textbooks” are neither professional nor helpful to your goal of publishing your work. For instance, the statement “According to standard textbooks, the OOP coercivity is expected to be negligible, regardless of the magnitude of the in-plane (IP) anisotropy, because the equilibrium magnetization lies within the plane” is technically inaccurate. Any differences in the strength of in-plane magnetic anisotropy must lead to measurable differences in coercivity when the magnetic field is applied out of plane (in the figure that the authors have decided to remove from the manuscript). That, too, is “textbook” knowledge and merits careful consideration.

Overall, I recommend giving the authors an opportunity to publish their work and letting readers judge the value of this study for themselves. The ambiguity in the report may spark follow-up investigations, which could be a positive outcome on its own.

Reviewer #2

(Remarks to the Author)

All the issues and concerns are addressed in the round of reply. I recommend for the publication.

Response to the Reviewers

Reviewer #1

Comment 1:

As demonstrated in [Physical Review B 90, 125311–6 (2014)], the proximity of C60 to cobalt can lead to the magnetization of the C60 layer through spin-polarized charge transfer from Co. Consequently, shortly after optical pumping, the C60 layer itself may become magnetized. This raises the possibility that the observed oscillations originate directly from the C60 layer rather than solely from the cobalt layer. Given the transient nature of the C60 magnetization, these oscillations would also be transient, which could explain the observed decaying behavior. The authors should explore this possibility in detail, supported by appropriate experiments or simulations, to provide a clearer description of the origin of the oscillations. A systematic variation in the thickness of the Co and/or C60 layers could offer valuable insights into the source of the oscillations and help clarify whether they arise from the cobalt layer, the C60 layer, or an interfacial effect.

Response to comment 1:

We appreciate the reviewer's suggestion and have carefully considered the possibility of the C60 layer contributing to the observed oscillations.

Firstly, we note that the study by Moorsom et al. (2014) reports static (XMCD) and quasi-static (SQUID) measurements, indicating a permanent spin-polarization of the C60 layer due to proximity effects, rather than a transient magnetization. Nonetheless, even if a transient magnetization of the C60 layer were present, our experimental design minimizes the C60 layer influence on the detected signal. Specifically, we chose the probe wavelength of 800 nm, where C60 is transparent, ensuring that the detected magneto-optical Kerr signal originates exclusively from the Co layer. This wavelength selection effectively rules out any direct contribution from potential magnetic dynamics within the C60 layer. Therefore, we attribute the observed oscillations in Kerr rotation to spin dynamics in the Co layer.

Furthermore, even if the probe were sensitive to magnetization within the C60 layer, we would expect the presence of additional magnetic modes from C60, coexisting with the known magnetic mode of the bulk Co layer. However, our experimental data does not show any such additional modes, further supporting the conclusion that the observed oscillations arise from the Co layer alone.

Finally, we acknowledge that the spin dynamics at the Co/C60 interface involve complex interactions requiring extensive experimental and theoretical exploration, which exceeds the scope of this paper. However, we address this complexity in two recent studies, now available on online repositories. One study focuses on the in-plane static magnetic properties of the Co/C60 interface [<https://doi.org/10.21203/rs.3.rs-4540787/v1>], while the other investigates GHz oscillations in time-resolved MOKE experiments in several Co/molecules systems [arXiv:2412.08677]. Both studies support the conclusion presented in this manuscript: the oscillations are localized within the Co layer and arise from the modification of its magnetic properties through interactions with the C60 layer. In particular, the second study [arXiv:2412.08677] rules out the possibility that the oscillations originate from the molecular layer. This conclusion is based on an extensive analysis of transient spin-wave dynamics in Co/molecule heterostructures featuring different molecular species. Our data indicate that upon hybridization with molecules, the Co/molecule hybridized interface develops

a strongly anisotropic magnetic state. This interfacial layer produces a large effective anisotropy field on the cobalt layer. In our experiments, we observe only a single precession frequency, which originates from the precession of the magnetization in the cobalt layer under the influence of this additional effective field induced by the Co/C60 interface. Notably, the magnetization of the interfacial layer adiabatically adapts to the magnetization of the Co layer during precession, further confirming that the observed oscillations stem from the Co layer rather than independent magnetization dynamics in the C60 or Co/C60 layer.

Changes to the manuscript:

- Added a reference to the Moorsom paper, relevant for our study.
- Expanded discussion on the experimental constraints that rule out a contribution from C60 layer magnetization.
- Strengthened the argument by referencing systematic studies (by some of the authors) on Co/molecule heterostructures: [<https://doi.org/10.21203/rs.3.rs-4540787/v1>] and [[arXiv:2412.08677](https://arxiv.org/abs/2412.08677)]

Comment 2:

Throughout the manuscript, the authors use general terms such as “oscillation,” “spin dynamics,” and “precession” without specifying the precise nature of the observed oscillations. Are these oscillations attributable to ferromagnetic resonance (FMR) modes of the cobalt layer, or do they correspond to spin waves? Clarifying this distinction is critical. For instance, could the observed differences between the Co/Al and Co/C60 samples be explained by a scenario where the Co/Al sample primarily exhibits FMR modes, while the Co/C60 sample supports high-frequency spin waves? This hypothesis might also account for the higher damping observed in the Co/C60 sample compared to Co/Al. The granular and semiperiodic structure of the C60 “soccer balls” at the top layer could act as a grating, providing the necessary k-matching condition to excite high-frequency spin waves at non-zero wave vectors. In contrast, this mechanism would be absent in the Co/Al sample, which might explain why only low-frequency FMR modes are accessible in that case. The authors should address these points and consider additional experiments or simulations to confirm the nature of the oscillations.

Response to Comment 2:

We thank the reviewer for raising these important points. While the literature [M. van Kampen et al., Phys. Rev. Lett. 88, 227201 (2002); J.-Y. Bigot et al., Chemical Physics 318, 137 (2005)] often refers to such coherent oscillations as “spin waves,” it is essential to clarify the nature of the observed oscillations in our system.

Our cobalt films are only 5 nm thick, significantly smaller than the typical magnetic exchange length (~10 nm). This means that higher-order standing spin-wave modes are pushed to much higher frequencies, leaving only the fundamental uniform FMR-like mode observable in our experiments. Additionally, any transverse k -mode that could be excited via a molecular grating effect would correspond to wavelengths on the order of a nanometer, leading to excitation frequencies on the scale of several hundred GHz or even THz. Given our experimental conditions, we detect only the dipolar ferromagnetic resonance (FMR) mode, corresponding to collective oscillations

of ferromagnetic (FM) spins with wave vector $k \sim 0$. Our approach—applying an external magnetic field to cant the magnetization out of the plane, followed by optical excitation to initiate magnetization precession around the effective field—is a well-established method for probing such FMR modes (see, e.g., Kampen, M et al., Ref. 23 in the revised manuscript).

We appreciate the reviewer's hypothesis regarding high-frequency spin waves and their potential role in explaining the differences between Co/Al and Co/C60 samples. However, we can confidently exclude this scenario based on the following considerations:

1. Frequency scale considerations:

The expected frequencies of finite- k spin waves in our system would be a significant fraction of the Co exchange-energy scale (~ 1000 T). These high frequencies are far beyond the GHz range observed in our experiments, making their contribution to the detected oscillations unlikely.

2. Sensitivity of the probe:

As stated in our response to Comment 1, our probe photon energy (800 nm) is not sensitive to magnetization dynamics within the C60 layer. Consequently, any spin waves originating directly from the C60 layer or driven by its structure cannot contribute to the detected signal.

3. Absence of additional modes:

If high-frequency standing spin waves were excited within any of the layers or at the Co/C60 interface, we would expect to observe multiple oscillatory components in our measurements, corresponding to both the fundamental FMR mode and the higher-order spin-wave modes. However, our experimental data consistently show a single dominant frequency, ruling out the presence of additional modes.

Based on these considerations, we conclude that the observed oscillations are indeed dipolar FMR modes arising from collective oscillations of the FM spins in the cobalt layer.

We acknowledge the reviewer's suggestion regarding additional simulations to further explore the magnetic dynamics at the Co/C60 interface. While a detailed microscopic study is beyond the scope of this paper, we have recently uploaded a comprehensive investigation of this system on the arXiv repository [arXiv:2412.08677]. As already mentioned in the reply to comment 1, in that study, we specifically address the hardening of the FMR mode frequency caused by the adsorption of C60 and other molecular layers and developed a model that successfully reproduces the magnetic field dependence of the observed FMR mode, further supporting our interpretation of the oscillations.

Changes to the manuscript:

- Revised the discussion in Section "Results" to explicitly state that the observed oscillations correspond to FMR modes of the Co/C60 system.
- Included references to our recent work on the arXiv [arXiv:2412.08677], which develops a comprehensive model of the Co/C60 interface and explains the observed magnetic dynamics.

Comment 3:

As demonstrated in [Phys. Rev. Lett. 88, 227201 (2002)], the optical excitation of magnetic oscillations typically occurs in multiple stages. Initially, light absorption leads to rapid heating of the magnetic material, which causes both demagnetization and changes in magnetic anisotropy, particularly shape-induced anisotropy. According to the referenced study (see Figure 1b), coherent magnetic oscillations only become observable after the heat is dissipated into the substrate, a process that takes several picoseconds, often up to ~10 ps. In contrast, the present manuscript reports the emergence of magnetic oscillations immediately following optical excitation, at pump-probe delays of approximately 0 picoseconds. Could the authors explain this discrepancy? It appears that the oscillations in the current study occur in a system that is thermally out of equilibrium. If this is indeed the case, one might expect the oscillation frequency to exhibit time dependence as the system cools down gradually. However, the reported oscillations seem to be time-independent. This apparent contradiction should be addressed and clarified in the manuscript.

Response to comment 3:

We appreciate the reviewer's observation and acknowledge the complexity of ultrafast magnetization dynamics in the first few picoseconds after optical excitation.

Given that the oscillation period is on the order of tens of picoseconds, determining the precise onset of oscillations with picosecond resolution is challenging. Several dynamic processes occur within the first few picoseconds following optical excitation. After the sub-picosecond demagnetization, the excess energy is first dissipated through electron-phonon relaxation (on a picosecond timescale), before significant heat removal to the substrate occurs. As a result, the magnetization dynamics in this early stage are complex and likely cannot be described as a simple sum of independent components.

However, based on our data, we can confidently state that the oscillations become clearly visible only after ultrafast demagnetization has occurred, the dominant part of electron-phonon thermalization has taken place (on a few-picosecond timescale), and the magnetization begins to recover. This recovery process occurs on a timescale of approximately 10 ps, which is in agreement with the scenario described by the reviewer. To clarify this point, we have added Figure S4d to the Supplementary Information (SI), showing the transient magnetization dynamics in the Co/C60 system during this recovery period.

The observed time delay before the oscillations become prominent reflects the time required for the effective magnetic field to realign and stabilize. This behavior is fully consistent with previous studies on ultrafast magnetization dynamics. Additionally, recent reports suggest that, under certain conditions, additional inertial spin dynamics such as nutations on the THz scale may also play a role in early-stage magnetization dynamics (e.g., Neeraj, K., Awari, N., Kovalev, S. et al., Nat. Phys. 17, 245–250 (2021), <https://doi.org/10.1038/s41567-020-01040-y>). However, these effects lie beyond the scope of the present study.

As previously stated in our response to Comment 2, our experimental setup is optimized to detect $k=0$ spin wave modes (FMR modes). The observed frequencies, as well as their dependence on external magnetic field and temperature, are consistent with the expected behavior of these modes, confirming the nature of the detected oscillations.

Figure S4d. *Example of time-resolved MOKE traces showing the pump-induced magnetization dynamics in Co/C60, highlighting the ~10 ps timescale during which ultrafast demagnetization and recovery occur.*

Changes to the manuscript:

- Added supplementary material: a zoomed-in view of a time-resolved MOKE trace on the ~10 ps timescale, illustrating the recovery of magnetization after ultrafast demagnetization (Figure S4d).
- Clarified the role of various processes in the initial stages of magnetization dynamics in the SI.

Comment 4.

The main claim is that excitons are modulating the magnetic anisotropy, and the change of anisotropy changes the oscillation frequency. Given the exponential decay of excitons density (after optical excitation), shouldn't we expect an exponential change of the oscillation frequency during the exciton lifetime?

Response to comment 4.

We thank the reviewer for pointing out this important aspect.

The referee is correct that the exciton lifetime (~150 ps) matches the timescale during which the $k=0$ spin waves can be detected. As the exciton density decays exponentially within this timeframe, we indeed expect a time-dependent oscillation frequency. However, analyzing this time dependence presents challenges due to the strong damping of the oscillations and the limited number of oscillation periods visible in our data.

To address this point, we conducted an extended analysis of our experimental data. Our findings reveal that the oscillation frequency is, in fact, time-dependent. The

frequency values reported in the main manuscript correspond to the early-time oscillations (i.e., within the initial period when the exciton density is still high and before significant decay occurs). For times comparable to the exciton lifetime, we observe deviations from a pure sinusoidal behavior, as shown in Figure S5, where the oscillations cannot be fully modeled by a single frequency.

It is important to note that this time-dependent frequency evolution is not in contradiction with the central claim of our manuscript—that the FMR frequency can be modified by selectively exciting the C60 molecules in the Co/C60 system. Rather, it highlights the transient nature of the exciton-driven modulation, which we now further elaborate on in the Supplementary Information.

Figure S5. Exemplary time-resolved MOKE curve from the Co/C60 system (450 nm excitation), illustrating that the oscillations cannot be fully described by a single frequency due to time-dependent modulation.

Changes to the manuscript:

- Added new material to the Supplementary Information (SI), providing an extended analysis of the time dependence of the oscillation frequency.
- Added a remark about the time-dependence of the oscillation frequency in the conclusions of the manuscript
- Updated references to this analysis in the main text to highlight the role of the exciton decay in modulating the FMR frequency.

Comment 5.

Page 2; Authors states: "...we investigate the potential for data storage, magnetic sensing, and quantum computing applications." I don't see any of these applications being investigated in this study. Can authors further elaborate what part of the manuscript this sentence is referring to?

Response to comment 5 and changes to the manuscript:

The referee is correct, we have modified the sentence accordingly. It now reads: "Inspired by advancements in molecular spintronics, which reveal molecules as a revolutionary platform for exploring spin-dependent phenomena at the nano-scale^{9,10}, we focus on harnessing proximity effects to modulate magnetic properties."

Comment 6.

Page 2; Authors write: “While ultrafast optical control has been demonstrated in dielectric materials⁷, metallic systems remain challenging due to rapid electron-hole pair dephasing and coherence loss caused by Coulomb screening⁸.” This statement does not sound accurate to me. A careful literature survey shows multiple examples of successful ultrafast and coherent optical control in metallic systems, see for instance [Science 382, 299–305 (2023)]. Authors need to conduct a better literature survey.

Response to comment 6:

We appreciate the reviewer’s feedback and have carefully reviewed the suggested reference. The cited work (Science 382, 299–305, 2023) focuses on the development of a technique for analyzing terahertz pulses emitted by hot carriers, which enables the study of spatiotemporal dynamics of hot-carrier transport in metals. However, this study does not demonstrate a coherent optical control scheme in the conventional sense.

As mentioned in the introduction of our manuscript, the dephasing time in metallic systems is on the order of a few femtoseconds due to rapid electron-hole pair dephasing and Coulomb screening. For coherent control to be feasible under these conditions, it would require operations on an attosecond timescale, a regime explored within the field of attosecond physics. While attosecond control is possible, it is outside the scope of our current work, which focuses on ultrafast dynamics in the femtosecond range.

Changes to the manuscript:

- We have clarified that our statement refers to coherent control on the femtosecond timescale, and we have revised the text accordingly to avoid ambiguity.

Comment 7:

The authors have primarily focused on the oscillation frequencies under various conditions (e.g., temperature, pump fluence) while largely neglecting the trends observed in the decay rates and, more critically, the amplitudes of the oscillations. A particularly concerning issue is the apparent lack of correlation between pump fluence and oscillation amplitude. For instance, as shown in Figure S3 for a 650 nm wavelength, increasing the pump fluence by a factor of four (from 0.016 to 0.071) results in only a minimal change in the oscillation amplitude. Similar discrepancies are observed in other measurements. One would typically expect a linear increase in amplitude with pump fluence, at least in the low-fluence regime, due to the direct relationship between absorbed energy and excitation strength. These observations raise questions about the underlying mechanisms and suggest that key aspects of the data have not been fully analyzed or discussed. The authors should carefully process and examine these trends in the revised manuscript, providing a detailed explanation of the observed behaviors and their implications.

Comment 8:

Following the above point; There is a very abstract but not conclusive discussion in the supporting information on decay rate with pump fluence. The trend for Co/C60 samples

is systematic at all tested wavelengths, showing a significant increase in the decay rate. For instance, at 650nm, the decay time drops from ~ 50ps to ~ 12ps as the pump fluence reaches 0.12mJ/cm². Interestingly a completely opposite trend is observed for the Co/Al sample. This seems too systematic to be ignored in the analysis of the modulation mechanism.

Response to comments 7 and 8:

We thank the reviewer for these insightful observations. The referee is correct that our initial manuscript primarily focused on the oscillation frequencies under various conditions while not discussing in detail the trends observed in the decay rates and oscillation amplitudes. The main reason for this was to maintain a clear focus on the achieved optical gating effect, as covering all aspects of the Co/C60 interface in a single manuscript would be impractical. However, we fully acknowledge that the data analysis can be improved to provide a clearer picture of the underlying trends.

We agree with the reviewer's expectation that, in the low excitation regime, the oscillation amplitude should show a linear dependence on the pump fluence. To address this, we have conducted a more refined data analysis, considering the maximal demagnetization peak (as shown in Figure S3) as an additional parameter to verify that the effective laser fluence impinging on the sample was correctly chosen. This allowed us to extract the frequency, amplitude, and decay rates with improved accuracy (Figures S4 and S5).

Our revised analysis leads to several key observations:

1. Optical Gating Effect Remains Valid:

The primary claim of our manuscript—that optical excitation of excitons in C60 modifies the FMR frequency—remains intact. The frequency dependence on fluence continues to support the role of exciton-mediated modulation.

2. Linear Dependence of Amplitude on Fluence:

The improved analysis now reveals a clear linear trend in the amplitude of the oscillations as a function of laser fluence, particularly for the 650 nm case, as expected in the low-excitation regime. This is shown in Figure S4a for the Co/C60 sample.

3. Distinct Trends in Decay Rates:

The extracted decay rates display a systematic trend in the Co/C60 sample, differing significantly from the behavior observed in the Co/Al reference sample (Figure S5). Specifically, for Co/C60, the decay time decreases as pump fluence increases, while an opposite trend is observed in the Co/Al sample. This systematic behavior, although not the primary focus of our manuscript, is indeed an intriguing aspect that warrants further investigation.

While a detailed exploration of these trends goes beyond the scope of this work, we highlight that the underlying mechanisms are addressed in recent studies. In particular, in the already mentioned preprint [arXiv:2412.08677], we analyze the hardening of k=0 spin wave modes induced by C60 adsorption on cobalt. Additionally, we direct the reviewer's attention to the other mentioned preprint by some of the co-authors [<https://doi.org/10.21203/rs.3.rs-4540787/v1>], which demonstrates how the formation of a hybrid interface between Co thin films and molecular layers leads to the

suppression of standard domain structures and the emergence of a glassy-like phase, driven by a correlated random anisotropy field. As pointed out by Chudnowsky et al. (*Eur. Phys. J. B*, 2024, <https://doi.org/10.1140/epjb/s10051-024-00825-x>), the presence of such a random anisotropy field can lead to a substantial increase in oscillation damping.

As the referee will recognize, this is a highly complex topic that, while partially understood, merits an independent study beyond the scope of the present manuscript. Given our primary focus on demonstrating the optical gating effect, we believe these additional investigations should be addressed in future work rather than included in the current manuscript.

Changes to the manuscript:

- Conducted an improved analysis of oscillation amplitude and decay rates, now presented in the SI (Figures S4b and S4c).
- Provided references to recent studies addressing the observed trends in damping and decay rates in the SI.

Figure S.4 Extracted parameters from tr-MOKE traces data for Co/Al (hollow points) and Co/C₆₀ (full points) samples. **a**, Pump-induced ultrafast magnetization peak. Extracted values of A_0 (**b**) and decay time τ_{DS} (**c**) for every pump wavelength used. **d**

tr-MOKE traces for the Co/C₆₀ sample showing the pump-induced ultrafast demagnetization on the Co/C₆₀ sample for a effective fluence of 0.045 mJ/cm².

Comment 9.

Page 4; referring to Figure 1, Authors express "...This set of measurements demonstrates that we can use the frequency of the GHz oscillations extracted from the tr-MOKE experiments as a measure of the anisotropy field ...". For a more quantitative calculation, authors can also leverage the field dependence of the oscillations and use the Standard Kittel formula for FMR frequency and back calculate the exact anisotropy field/energy of Co/C₆₀. Is there any reason this has not been done here?

Response to comment 9:

We appreciate the reviewer's suggestion and acknowledge the value of quantitatively extracting the anisotropy field using the standard Kittel formula. However, as described in our related preprint [arXiv:2412.08677], the microscopic details of the hybridized Co/molecule interface extend beyond the framework of a simple quadratic anisotropy model.

Specifically, the frequency dependence on the out-of-plane external magnetic field observed in our experiments is not compatible with standard FMR formulas based on a purely quadratic anisotropy term. Moreover, this behavior appears to contradict the static out-of-plane magnetization measurements, indicating that a more complex anisotropy landscape is at play. A full theoretical description of both the in-plane and out-of-plane static magnetization together with the dynamical FMR response requires a more advanced modeling approach.

Our study of multiple molecular species [arXiv:2412.08677] indicates that upon hybridization with molecules, the interfacial Co layer develops a strongly anisotropic in-plane magnetic state. Due to its fast dynamics, this state influences both the static magnetization and the FMR response. This effect can be effectively described using an in-plane exchange-bias-like free energy term in combination with a reduction of the standard out-of-plane hard-axis anisotropy.

The present manuscript focuses on the modulation of this effective anisotropy by laser excitation. While a detailed theoretical treatment of the anisotropy is crucial for a deeper understanding, including such an extensive discussion in this paper would significantly increase its complexity and make it less accessible to a broader audience. Therefore, we choose to describe the anisotropy in terms of the effective anisotropy

field extracted directly from our measurements, avoiding reliance on additional assumptions.

Changes to the manuscript

- Added a brief description of the origin of the anisotropy.
- Included a reference to [arXiv:2412.08677] to direct interested readers to a more detailed discussion of the anisotropy landscape in hybrid Co/molecule interfaces.

Comment 10.

Page 9; Authors write "... we observe the modification of the precession frequency at $wCo = 0.1$ mJ/cm². At this energy density, we note an extremely significant modulation of more than 60% between the resonant and off-resonant pumping". However, It's not clear how the 60% modulation is calculated. The resonant pumping is 450nm wavelength condition. Looking at Figure 2b, no data point is provided at 0.1 mJ/cm² for this wavelength.

Response to comment 10:

We thank the reviewer for outlining this point. The fluence to which we refer to is 0.07 mJ/cm². We changed the manuscript accordingly. The 60% is modulated by calculating:

$$1 - \frac{\nu_{CoC60}(650 \text{ nm}) - \nu_{CoC60}(450 \text{ nm})}{\nu_{CoC60}(650 \text{ nm})}$$

Changes to the manuscript:

- Mistake was corrected.

Comment 11.

Figure 2a; Authors should include the absorption spectra of the pure C60 layer with the same thickness as the one used on Co/C60 experiments? This will enable a better understanding of the optical response of the C60 independent of the interfacial effects from the Co layer.

Response to comment 11.

We appreciate the reviewer's suggestion. The optical absorption spectrum of bulk C60 is well-documented in the literature, and we have cited the relevant studies in our manuscript. Our findings align perfectly with these established spectra, confirming the expected optical response of the C60 layer. Given this agreement and the extensive availability of reference data, we do not consider it necessary to perform additional absorption measurements for a pure C60 layer of the same thickness.

Comment 12.

Figure 2a; can authors comment on the step-like jump in the optical absorption obtained from Co/Al in the wavelength range between 550-600nm?

Response to comment 12:

We thank the reviewer for indicating this. The Co/Al sample was measured some months after the tr-MOKE characterization. In order to be more consistent, we repeated the measurement on a fresh Co/Al sample, and such shoulder is disappeared. We can infer that there was either a measurement artifact or a degradation of the Co/Al sample over time.

Changes to the manuscript

- Replaced Fig. 2a with a new dataset

Comment 13.

Figure 1; Figures 1c and 1d show that the oscillation frequency of the Co/C60 sample is significantly higher than that of the Co/Al sample, differing by a factor of approximately 5–6 at low temperatures. The authors attribute this difference to the stronger anisotropy in the Co/C60 sample compared to Co/Al. However, this substantial difference in anisotropy is not reflected in the magnetization curves presented in Figure 1b, where both samples reach their saturation values at nearly the same magnetic field. This observation is counterintuitive, as the saturation field is directly influenced by the anisotropy field, which should differ significantly between the two samples if the stated explanation holds.

Response to comment 13:

We appreciate the reviewer's observation and agree that a more detailed discussion is warranted. The hysteresis loops presented in Figure 1b are affected by a relatively high noise level, which may obscure subtle differences in the magnetization behavior of the two samples. However, as discussed in our response to Comment 9 and in our related study [arXiv:2412.08677], there exists an apparent dichotomy between the out-of-plane static magnetization and the coherent FMR response.

Our findings indicate that upon hybridization with molecules, the interfacial Co layer develops a strongly anisotropic in-plane magnetic state. Due to its fast dynamics, this state influences both the static and dynamic magnetic properties of the system. Specifically, its effect on the static out-of-plane magnetization is relatively minor, as the in-plane interfacial anisotropy does not significantly alter the saturation field in Figure 1b. However, its influence on the FMR dynamics is much more pronounced. The strong frequency hardening observed in Figures 1c and 1d can be effectively captured by describing the interfacial contribution as an in-plane exchange-bias-like free energy term combined with a reduction in the standard out-of-plane hard-axis anisotropy. This unique combination leads to only subtle changes in the static magnetization curves while simultaneously driving a significant increase in the coherent FMR frequency.

Thus, while the saturation behavior in Figure 1b may not immediately suggest the presence of a substantially increased anisotropy, the interplay between interfacial anisotropy and exchange coupling leads to the observed frequency shift in the Co/C60 sample.

Changes to the manuscript:

- Clarified the apparent discrepancy between static magnetization and FMR frequency hardening.
- Referenced [arXiv:2412.08677] to provide further context on the role of interfacial anisotropy.
- Explained how an in-plane exchange-bias-like term can modify the FMR response without drastically affecting static magnetization curves.

Comment 14:

The central focus of the manuscript is on magnetic anisotropy and its modulation for technological applications. However, none of the cited works adequately highlight the significance of magnetic anisotropy in spintronic applications. Including references to studies where anisotropy serves as a critical enabler for Spintronics could strengthen the manuscript. Here are some examples of the critical role of magnetic anisotropy in Data storage [Adv. Mater. Technol. 2023,8,2300676], Field-free magnonics [arXiv:2411.14428, <https://doi.org/10.48550/arXiv.2411.14428>], and neuromorphic computing [Phys. Rev. Applied 19, 064018 – Published 6 June, 2023].

Response to comment 14 and changes to the manuscript:

Relevant literature was included.

Reviewer #2

Comment 1:

The authors have analyzed the temperature-dependent oscillation frequency variations (Fig.1a) and the power-dependent frequency changes (Fig.2b), the experimental phenomena are clear. However, in the analysis of the temperature dependence, the authors have only provided a qualitative explanation of the observed experimental phenomena without offering a detailed physical interpretation of these observations. It is only stated that at low temperatures, the data reveal an approximately five-fold increase of ν , which agree with the literature's observations of an enhanced Hanis value. This enhancement is attributed to the hybridization of electronic states at the Co/C60 interface, which significantly alters the magnetic anisotropy. This statement is not sufficient for the audience to fully understand the physics behind the observations.

Comment 2:

The author mentions that GHz oscillation frequencies can be utilized to measure the anisotropy field (Line108), yet does not provide further details or implementation strategies. I believe the author could enhance the discussion on this point.

Response to comments 1 and 2:

We appreciate the reviewer's insightful comment and agree that a more detailed physical explanation would enhance the clarity of our manuscript.

However, as we also explained in the reply to referee 1, the underlying physics governing the temperature dependence of the oscillation frequency is quite complex and extends beyond a simple quadratic anisotropy model. This topic has been extensively addressed in our related study [arXiv:2412.08677]. Our findings reveal an apparent dichotomy between the static out-of-plane magnetization and the dynamic FMR response. Specifically, upon hybridization with molecules, the interfacial Co layer develops a strongly anisotropic in-plane magnetic state. Due to its fast dynamics, this state influences both the static and dynamic properties of the system. While its effect on the static out-of-plane magnetization remains relatively minor, its impact on the coherent FMR response is significant.

This behavior can be effectively described using an in-plane exchange-bias-like free energy term in combination with a reduction of the standard out-of-plane hard-axis anisotropy. The present manuscript focuses on the modulation of this effective

anisotropy by laser excitation. While a full theoretical treatment of the anisotropy would provide deeper insights, including such a detailed discussion here would significantly increase the manuscript's complexity and reduce its accessibility to a broader audience. Therefore, we choose to describe the anisotropy in terms of the effective anisotropy field extracted directly from our measurements, without relying on additional assumptions.

*This approach is well-established and widely utilized in FMR spectroscopy, as discussed in standard textbooks (see, for example, S. V. V. Sovskii, *Ferromagnetic Resonance: The Phenomenon of Resonant Absorption of a High-Frequency Magnetic Field in Ferromagnetic Substances*, Pergamon Press Ltd., Oxford, 1966). In the context of TR-MOKE experiments, the use of GHz oscillation frequencies as a measure of anisotropy fields was discussed, for example, in [J.-Y. Bigot et al., *Ultrafast magnetization dynamics in ferromagnetic cobalt: The role of the anisotropy*, *Chemical Physics* 318, 137 (2005)].*

Changes to the manuscript:

- Expanded the discussion in Section "Results" to include a summary of the more detailed physical and theoretical interpretation of the experimental results from our related manuscript [arXiv:2412.08677].
- Added references to standard FMR literature and prior TR-MOKE studies to guide readers less familiar with the background on how GHz oscillations can be used to determine anisotropy fields.

Comment 3:

In the present case, 25-nm-thick C60 film was used with the exposure of external field of $H_{ext}=0.5T$. How the thickness of the c60 layer and the external magnetic field on the functionality of the hybrid system, for example, on the modulation of the spin precession frequency?

Reply to comment 3

We thank the reviewer for raising this important point.

First, we clarify that the external magnetic field $H_{ext}=0.5 T$ is applied solely to facilitate the detection of the $k=0$ spin wave mode (FMR mode) in our optical measurements. This is a necessary experimental condition for ensuring a well-defined resonance mode, rather than a parameter directly influencing the gating mechanism. We have revised the manuscript to make this point clearer.

Regarding the thickness of the C60 layer, its primary role is to ensure full and homogeneous coverage of the Co layer, rather than actively influencing the spin precession frequency. Previous studies [Cinchetti 2016, 2017; Moorsom 2020] have

shown that only the first molecular layer—and to some extent, the second—participates in the formation of a hybrid interface with the transition metal. Specifically:

- The first C60 monolayer undergoes hybridization, leading to a broadening and spin-splitting of its molecular orbitals.
- The second monolayer can develop interface-induced electronic states with a spin-dependent lifetime [Cinchetti & Droghetti 2016].
- Beyond the second monolayer, additional C60 layers do not significantly contribute to the interfacial electronic or magnetic properties.

Despite these theoretical considerations, we experimentally verified the impact of Co thickness by measuring samples with different Co layer thicknesses before performing the gating experiments presented in the manuscript. These measurements allowed us to determine the optimal Co thickness for maximizing the Kerr response. The results of this thickness-dependent study are now included in the Supplementary Information (Figure S8).

Figure S8. Example of in-plane hysteresis loops of Co/C60 at 80 K, showing a quantitative increase of the Kerr rotation with increased Co thickness.

Changes to the manuscript:

- Clarified the general measurement scheme that allows detecting the FMR mode in TR-MOKE experiments and the necessity to apply an external magnetic field.
- Added the new dataset (Figure S8).

Comment 4

In Fig2a, three Gaussian functions are used to fit the broad Co/C60 absorbance spectrum (390nm-550nm), however, only the fitted curve (in yellow) is presented in the inset. I recommend displaying the fitted results alongside the measured data, similar to the approach taken in Fig. S6. Furthermore, I highly recommend replotting Fig. S6(a-g) to ensure that the measured data is distinctly discernible from the fitted curve. In its current presentation, the measured data (represented by circles) is barely visible, as it is overshadowed by the "thick" solid lines of the fitted data. The label for the x-axis of Fig.S6h is missing.

Response to comment 4 and changes to the manuscript

We thank the referee for pointing out these problems and have solved all of them in the revised version.

Comment 5.

Why is the pump fluence in Fig. 2b only approximately 0.15 mJ/cm² for the Co/C60 sample, while it reaches around 0.35 mJ/cm² for the Co/Al sample? Within the range of 0.15 mJ/cm², the precession frequency ν changes linearly with fluence. Is this linear relationship maintained beyond the 0.15 mJ/cm² threshold, and are there corresponding experimental measurements to support this?

Response to comment 5

We appreciate the reviewer's observation and agree that further clarification is needed regarding the fluence range used for the Co/C60 sample in Figure 2b.

Our analysis for Co/C60 was intentionally limited to low fluences (~0.15 mJ/cm²) because, beyond a certain threshold, the observed time-resolved MOKE (TR-MOKE) traces could no longer be accurately modeled using a simple exponential decay plus a damped sinusoidal function. At higher fluences, the Co/C60 system exhibits a more complex behavior, suggesting the presence of additional dynamical effects beyond the scope of this study. Our goal in this work is to demonstrate the tuning of magnetization dynamics at relatively low fluences, which is, in itself, a significant achievement, as it implies efficient optical control without requiring high-energy excitation.

Additionally, we note that the effective fluence reaching the Co layer depends on the absorption by the C60 layer, which is different from the Co/Al sample. The pump power was adjusted accordingly and converted into fluence, leading to a different fluence range for Co/C60 compared to Co/Al.

To illustrate the deviation from simple oscillatory behavior at high fluences, we now provide an example TR-MOKE trace in the Supplementary Information (more can be seen in Fig. S6 and S7 of the revised SI), demonstrating the non-trivial response of the Co/C60 system at higher fluences.

Furthermore, as part of our response to Reviewer #1 (Comment 7), we have performed a more refined analysis of the fluence-dependent data. This improved analysis confirms the expected linear relationship between oscillation amplitude and fluence in the low-fluence regime.

Example of a tr-MOKE trace for the Co/C60 sample at the highest achievable fluence at 2.75 eV pump photon energy.

Changes to the manuscript:

- Clarified the reason for limiting the analysis to low fluences for Co/C60.
- Added Figure S6 in the Supplementary Information, showing an example of a high-fluence TR-MOKE trace for Co/C60 to illustrate the onset of complex dynamics.

Comment 6.

The abbreviation “CT” should be defined in an appropriate position, i.e., in Line 207 where “CT” is for the first time appears: “...lower-lying CT states...”.

Response to comment 6 and changes to the manuscript.

Done

Comment 7.

Issue of figure arrangements: honestly, I do not like the presentation of the figures in the present manuscript: The authors have not arranged the figures with diligence, resulting in low resolution images where many symbols and fonts are difficult to distinguish. For example:

In Fig.1b: the label of x-axis is hardly visible.

In Fig.1d, the error bar of the last value of the precession frequency is missing.

8. Line82 : There is an extra "the" in line 82.

9. Line378: In Ref.17, article number is missing

10. Line382: In Ref.18, a dash is missing in the page number

11.Line406: In Ref.27, a dash is missing in the page number

Response to comment 7:

We appreciate this feedback and have revised the figures accordingly. We have improved the resolution of all figures, adjusted axis labels for better readability, and included the missing error bars in Fig. 1d.

Changes to the manuscript:

- Improved resolution and formatting of all figures.
- Added missing error bars and adjusted axis labels.

Reviewer #3

Comment 1:

The abstract needs to be substantially beefed up to present better the findings of the authors' study. As it is, its content is vague.

Response to comment 1 and changes to the manuscript:

We have rewritten the abstract to more clearly highlight our key findings and their significance. The new abstract describes the impact of our results on the understanding of proximity-enhanced functionalities, while keeping the word limit of 200 words imposed by Nature Comm.

Comment 2:

The conclusions paragraph in the final part of the article is rather dry and does not provide the reader with any perspective on future usage of the authors' findings. I encourage the authors to rework this final part and emphasize the technologically relevant aspects of their work.

Response to comment 2 and changes to the manuscript:

We have expanded the conclusions section to discuss potential applications of our work in magnetic sensing and spintronics. We also added references to studies where magnetic anisotropy plays a key role in technological applications.

Reply to reviewers

Reviewer #1 „Unresolved Fundamental Questions“:

(Questions 1, 2 and 3)

(1) The authors claim that their new study on arXiv [arXiv:2412.08677] confirms that the oscillations stem from the ferromagnetic resonance (FMR) of Co rather than the magnetized C60 layer. However, Figure 1b contradicts this claim. The nearly identical coercive fields of Co/C60 and Co/Al suggest that hybridization of C60 with Co does not significantly alter the anisotropy, contrary to what the authors imply. Their explanation involving "complex anisotropy terms" and an "exchange-bias anisotropy" remains vague and lacks proper elaboration in the manuscript. Given that this aspect underpins the entire discussion of light-induced anisotropy modulation, a clearer and more rigorous analysis is necessary.

(2) The Kerr rotation at saturation in the Co/C60 sample is almost twice that of the Co/Al sample (Figure 1b), despite the Co layer thickness being identical in both cases. This strongly suggests that the probe light is detecting contributions from magnetized C60, contradicting the authors' assertion that the Kerr signal originates solely from Co. Their response—stating that C60 is transparent at 800 nm—is not convincing because it does not account for the fact that an out-of-equilibrium C60 layer (excited by the pump pulse) could have altered optical properties. The authors should explore this possibility rather than dismissing it outright.

(3) Overreliance on the arXiv Manuscript: A key part of this study's analysis appears to be conducted and justified elsewhere, in a separate arXiv manuscript. This raises concerns about the independence of the current work and whether it stands on its own merit. The authors frequently defer critical discussions to another paper on arXiv, stating that this arxiv paper (1) proves that the FMR stems from Co rather than C60, (2) explains why Co/C60 and Co/Al have similar coercivity and anisotropy in Figure 1b, and (3) describes quantitative values of the anisotropy and Kittel behavior of the FMR. If these essential aspects are primarily addressed elsewhere, then what exactly is the novel contribution of the present manuscript? Simply demonstrating that FMR frequency shifts upon laser excitation is not sufficient to warrant publication. Moreover, other key aspects, such as trends in the decay rate and oscillation amplitudes, remain largely unexplored, with the justification that the system is too complex. This significantly weakens the study's impact.

Reply to „unresolved fundamental questions“

Based on these questions we understand that the manuscript needs to contain a more rigorous description of the physics observed and be as self-contained as possible, without relying too much on other manuscripts that some of us have prepared on the same system. For this reason, in the newly revised version we explain more in detail the physics related to the effects of C₆₀ on the cobalt substrate.

Before describing in detail the changes made in the manuscript to address this crucial point (see "Changes to the manuscript" below), we would like to point out that the Co/C₆₀ system is extremely complex, and that the two manuscripts we were referring in the original reply deal with the characterization of the static magnetic properties of the system, including the properties of the k=0 (dipolar) spin waves. The present manuscript, on the other hand, starts from this knowledge and makes a significant step forward, by showing that creating excitons in the C₆₀ layer it is possible to quench the hybridization of the C₆₀ and thus to modulate the properties of the Co/C₆₀ system using light. As was already recognized by the referees in the previous revision round this is an exceptional discovery, because without using the C₆₀ as an active, optically responsive element, and the proximity coupling of C₆₀ to Cobalt, it would be impossible to achieve such a strong and deterministic optical modulation of the spin wave frequency (and the related anisotropy field) in the cobalt layer on the femtosecond-to-picosecond time scale.

Changes to the manuscript

In order to make the manuscript more self-contained, we have added the description of the Co/C₆₀ system in the manuscript and added an explanatory figure (**Figure 1a**). The added text reads:

At low temperatures, the Co/C₆₀ system shows an enhancement in ν by nearly a factor of five, indicating a substantial increase in the anisotropy field induced by the proximity of C₆₀. This observation is consistent with previous reports of enhanced H_{anis} values^{17,25}, attributed to hybridization between the Co electronic states and the C₆₀ molecular orbitals^{11,13}, which substantially modifies the interfacial magnetic anisotropy.

To place these findings in a broader context, we note that the observed proximity-induced modulation of spin dynamics is rooted in a universal interfacial mechanism recently identified across multiple cobalt/molecule heterostructures^{17 28}. In that study, a systematic TR-MOKE investigation revealed the emergence of a strongly anisotropic interfacial magnetic layer, which is driven by chemical hybridization between Co surface 3d-orbitals and molecular π -systems. The resulting interfacial layer, with magnetization \mathbf{m} (**Figure 1a**), has been recently shown to be fundamentally different from the magnetic

structure of the underlying magnetic layer with magnetization \mathbf{M} and a conventional uniaxial magnetic anisotropy. The formation of this interface dramatically modifies the magnetic state of the whole ferromagnet, leading to the emergence of a Correlated Ferromagnetic Glass¹⁷, with a correlated random anisotropy term K_R induced by surface hybridization with the molecular orbitals.

To describe the dynamics of such state in the ps timescale, the following free energy functional was recently proposed²⁸:

$$F = \mu_0 \xi_C^2 \sum_{\beta} \frac{(\nabla M_{\beta})^2}{2} - \mu_0 \mathbf{H} \cdot \mathbf{M} + \frac{K_{\perp}}{2} (\mathbf{M} \cdot \mathbf{e}_{\perp})^2 - K_R \left| \frac{\mathbf{M}}{M_0} \cdot \mathbf{e}_R \right|^{\alpha}$$

The first three terms correspond to exchange, Zeeman, and out-of-plane anisotropy energies. The last term is a phenomenological expression capturing the effect of the surface hybridization on the cobalt layers. It contains the anisotropy energy density parameter K_R whose easy local direction \mathbf{e}_R is random but correlated over a lengthscale defined by a correlation radius r_C . This term governs the anisotropy field around which the magnetization \mathbf{M} , coupled to \mathbf{m} via exchange interaction (\mathbf{J} in **Figure 1a**), precesses in our experiments. Due to the coupling, the interface layer follows the precession of \mathbf{M} adiabatically²⁸. As a result, the precession frequencies extracted in our experiments are directly sensitive on the parameters K_{\perp} , K_R and \mathbf{e}_R , and thus provide a means to monitor the interfacial modifications induced by C_{60} hybridization.

Building on this framework, our work demonstrates for the first time that interfacial hybridization—as captured by K_{\perp} , K_R and \mathbf{e}_R —can be dynamically controlled via resonant optical excitation. We show that it is possible to optically quench the Co– C_{60} hybridization by resonant exciton formation in C_{60} , and as a result to modulate the correlated random anisotropy field on picosecond timescales. As schematically illustrated in **Figure 1a**, this capability introduces a new degree of control over proximity-induced magnetic phenomena, with promising implications for the development of ultrafast spintronic devices.

Detailed answers:

Question 1. The hysteresis curves in **Figure 1b** were measured using the polar MOKE geometry, which probes the *out-of-plane* (OOP) component of the magnetization. In this geometry, the magnetic field is applied perpendicular to the film plane (i.e., to the easy plane), and the hysteresis reflects the response along the hard axis. According to standard textbooks, the OOP coercivity is expected to be negligible, regardless of the magnitude of the *in-plane* (IP) anisotropy, because the equilibrium magnetization lies within the plane. The observed near-zero OOP coercivity is therefore entirely consistent with expectations. The purpose of showing this hysteresis was simply to confirm the OOP direction as a hard axis; changes in IP anisotropy induced by C₆₀ adsorption are not captured in this measurement.

Changes to the manuscript: Based on the above reasoning, we believe that objection (1) arises from a misunderstanding of the measurement geometry. Since the presented hysteresis seems to have caused confusion, we have moved it to the Supplementary Information. In the main text, we now show a schematic of the measurement geometry along with the relevant physical context. We have also added in-plane hysteresis loops for Co(5nm)/C₆₀ and Co(5nm)/Al(3nm) in the Supplementary to show the magnetic hardening induced by the Co/C₆₀ interface.

Question 2.

We would like to make three important points:

1. **Hybridized interfacial layer:** As explained in the general response, our model of the Co/C₆₀ interface explicitly includes a strongly hybridized interfacial Co/C₆₀ layer, characterized by significantly enhanced in-plane random anisotropy. Crucially, this interfacial layer governs the dynamical response of the entire system—that is, both the hybridized top layer and the underlying cobalt layers that are exchange-coupled to it. We therefore do not disregard the role of the hybridized C₆₀ molecules; on the contrary, they are central to the system's magnetic and dynamical properties. In contrast, C₆₀ molecules located farther from the Co surface are not hybridized and are effectively transparent to the probe wavelength used in the experiment.
2. **Transient reflectivity:** Regarding the transient optical signal, we note that the reflectivity change measured in C₆₀/Co (**Fig. S11**) remains below 1%, indicating that significant photoinduced modifications of the optical properties of the non-hybridized C₆₀ layer are not supported by the experimental data.
3. **Kerr signal interpretation:** It is a well-known fact (see, e.g., Ahn and Fan [K. Ahn and G. Fan, *IEEE Trans. Magn.* **2**, 678 (1966)]) that the presence of a dielectric overlayer can strongly modulate the Kerr angle. Therefore, directly comparing Kerr

signal magnitudes across different heterostructures to infer or compare saturation magnetizations is fundamentally flawed and physically unjustified.

Question 3. In the paragraph added in the revised version, we have clarified the content of the Arxiv manuscript, that provide a solid theoretical and experimental basis to understand the static magnetic properties of the Co/C₆₀ interface. The focus of our manuscript is on the dynamical properties, and thus completely different from the focus of the Arxiv paper, which is now also clearly schematized by the newly introduced figure 1a. Furthermore, we have revised the introduction to more clearly describe the impact of C₆₀ hybridization on cobalt's magnetic properties and to delineate the novel aspect of our work: the active, light-induced control of the anisotropy field in the GHz regime.

Question 4

Overstatement of Technological Relevance: There is excessive use of promotional language throughout the manuscript, with claims of "novelty," "groundbreaking methods," and enabling "next-generation qICT technologies" and a "new paradigm in spin-based information processing." I find the claims very exaggerating; The described technique requires femtosecond optical pulses, operates at cryogenic temperatures, and necessitates high magnetic fields (~ 0.5T) none of which are welcomed in technology. Also, I don't see any "new paradigm" proposed or demonstrated here. Modulating magnetic properties with optical pulses are well-established. The use of a new material platform in this "established" paradigm does not mean there is a "new paradigm". The paper's focus should be on the fundamental physics of the platform, rather than prematurely marketing it as a technological breakthrough. If the authors wish to emphasize technological relevance, they must provide supporting data on power consumption, device footprint, scalability, and efficiency. Otherwise, the manuscript should be revised to focus on the fundamental physics and allow readers to assess the novelty of the findings.

Reply: we agree that it is not the method to be groundbreaking but instead the new material platform, i.e. the use of molecules and proximity effects to achieve optical control of magnetic properties.

Changes in the manuscript:

We reworded the abstract and introduction accordingly.

Other minor comments:

- **Minor comment 1.** The figure caption states a 400 nm pump, while the main text refers to 350 nm. This inconsistency needs clarification.

Reply: We have corrected the caption with the correct value (351 nm).

*- **Minor comment 2:** The authors attribute the step-like behavior in their data to sample degradation. However, this raises concerns about whether such degradation could also affect the pump-probe measurements. A discussion on this possibility is warranted.*

Reply: As we clarified in the first revision round, the original absorption spectrum (showing the mentioned step) was measured a few months after the TR-MOKE experiments, which on the other hand were performed on freshly prepared samples.

For this reason, in the first revision, we repeated the absorption measurements on freshly prepared samples, comparable to those studied with TR-MOKE. Here, the step is not visible, thus confirming that it was a sign of degradation. This is irrelevant for the measurements reported in the manuscript since they were performed on freshly prepared samples.

*- **Minor comment 3:** A systematic experiment could involve inserting a conductive layer (e.g., Cu) between C60 and Co. A Cu interlayer would allow charge transfer while blocking interfacial hybridization, providing further insight into the role of C60. The authors should consider such an experiment to strengthen their claims.*

Reply: In the revised version, we have provided a fully consistent explanation of the experiments presented in the manuscript, as well as of the nature of the magnetism at the Co/C₆₀ interface. We hope that the manuscript will motivate our groups to perform further measurements, but we think the manuscript in its present form is self-contained enough to warrant publication in its present form.

Reply to reviewers

We thank the reviewers for their careful re-evaluation of our manuscript and for acknowledging the substantial improvements in the revised version.

To address their final critique—and as also requested by yourself—we have now added a dedicated comment in the Discussion section of the revised manuscript to explicitly propose future experimental tests that could further substantiate our conclusions. Specifically, we now write:

"To further substantiate the optical tunability of proximity-enhanced functionalities in hybrid systems, future experiments could explore the role of interfacial hybridization more systematically by introducing spacer layers with varying electronic coupling strength between C_{60} and the ferromagnetic substrate. These control measurements would allow disentangling proximity-induced effects from other possible contributions. While such studies go beyond the scope of the present work, they represent a compelling direction for validating and generalizing the proposed mechanism across a broader class of molecule/metal interfaces."

Additionally, as also suggested by the author guidance document, we have revised the wording in various passages—replacing terms such as striking and novel—to adopt a more neutral and balanced tone.

Report for NCOMMS-24-70056-T

“Light-driven modulation of proximity-enhanced functionalities in hybrid 1 nano-scale systems”

This study demonstrates the optical functionality of Co films in close proximity to C60 molecules, offering an approach for manipulating spin dynamics. By using resonant ultrashort light pulses to induce excitons in C60, the authors show a significant alteration in the hybridization between the Co layer and the molecules. This process enables the modulation of GHz spin dynamics in the Co layer, resulting in up to a 60% change in the precession frequency, which is associated with a modification of the anisotropy field.

After thoroughly reviewing the manuscript, I find that I cannot recommend it for publication in its current form. My assessment is primarily based on two aspects: the theoretical explanation and the arrangement of figures. The authors need to address the following issues:

1. The authors have analyzed the temperature-dependent oscillation frequency variations (Fig.1a) and the power-dependent frequency changes (Fig.2b), the experimental phenomena are clear. However, in the analysis of the temperature dependence, the authors have only provided a qualitative explanation of the observed experimental phenomena without offering a detailed physical interpretation of these observations. It is only stated that at low temperatures, the data reveal an approximately five-fold increase of ν , which agree with the literature's observations of an enhanced H_{anis} value. This enhancement is attributed to the hybridization of electronic states at the Co/C60 interface, which significantly alters the magnetic anisotropy. This statement is not sufficient for the audience to fully understand the physics behind the observations.
2. The author mentions that GHz oscillation frequencies can be utilized to measure the anisotropy field (Line108), yet does not provide further details or implementation strategies. I believe the author could enhance the discussion on this point.
3. In the present case, 25-nm-thick C60 film was used with the exposure of external field of $H_{\text{ext}}=0.5T$. How the thickness of the c60 layer and the external magnetic field on the functionality of the hybrid system, for example, on the modulation of the spin precession frequency?
4. In Fig2a, three Gaussian functions are used to fit the broad Co/C60 absorbance spectrum (390nm-550nm), however, only the fitted curve (in yellow) is presented in the inset. I recommend displaying the fitted results alongside the measured data, similar to the approach taken in Fig. S6. Furthermore, I highly recommend replotting Fig. S6(a-g) to ensure that the measured data is distinctly discernible from the fitted curve. In its current presentation, the measured data (represented by circles) is barely visible, as it is overshadowed by the "thick" solid lines of the fitted data. The label for the x-axis of Fig.S6h is missing.
5. Why is the pump fluence in Fig. 2b only approximately 0.15 mJ/cm^2 for the Co/C60 sample, while it reaches around 0.35 mJ/cm^2 for the Co/Al sample? Within the range of 0.15 mJ/cm^2 , the precession frequency ν changes linearly with fluence. Is this linear relationship maintained beyond

the 0.15 mJ/cm² threshold, and are there corresponding experimental measurements to support this?

6. The abbreviation "CT" should be defined in an appropriate position, i.e., in Line 207 where "CT" is for the first time appears: "...lower-lying CT states...".

7. Issue of figure arrangements: honestly, I do not like the presentation of the figures in the present manuscript: The authors have not arranged the figures with diligence, resulting in low resolution images where many symbols and fonts are difficult to distinguish. For example:

In Fig.1b: the label of x-axis is hardly visible.

In Fig.1d, the error bar of the last value of the precession frequency is missing.

8. Line82: There is an extra "the" in line 82.

9. Line378: In Ref.17, article number is missing

10. Line382: In Ref.18, a dash is missing in the page number

11.Line406: In Ref.27, a dash is missing in the page number

I have carefully read the authors' response letter multiple times, and I appreciate the effort they have put into responding to my comments and revising the manuscript. However, I remain unconvinced by their responses, as they largely avoid providing direct and substantive answers. The authors frequently argue that (1) the material platform is too complex to draw definitive conclusions, (2) the signals are too weak to extract the necessary data, or (3) the raised concerns are addressed in a separate paper recently uploaded to arXiv. While these points are understandable, they ultimately leave the current manuscript as little more than an (interesting) observation, lacking the depth and details necessary to advance the field or provide a clear understanding of the underlying physics. I also note that Reviewer 2 has raised similar concerns. As such, I cannot recommend the publication of this manuscript in its current form. However, I strongly encourage the authors to address the following critical points, some of which were raised in my initial feedback:

- **Unresolved Fundamental Questions:** There are still key unresolved issues, particularly the origin of the observed oscillations and the exact nature of the anisotropy being modulated by the pump laser. The responses provided do not offer a satisfactory explanation for these essential points. Specifically: (1) The authors claim that their new study on arXiv [arXiv:2412.08677] confirms that the oscillations stem from the ferromagnetic resonance (FMR) of Co rather than the magnetized C60 layer. However, Figure 1b contradicts this claim. The nearly identical coercive fields of Co/C60 and Co/Al suggest that hybridization of C60 with Co does not significantly alter the anisotropy, contrary to what the authors imply. Their explanation involving "complex anisotropy terms" and an "exchange-bias anisotropy" remains vague and lacks proper elaboration in the manuscript. Given that this aspect underpins the entire discussion of light-induced anisotropy modulation, a clearer and more rigorous analysis is necessary. (2) The Kerr rotation at saturation in the Co/C60 sample is almost twice that of the Co/Al sample (Figure 1b), despite the Co layer thickness being identical in both cases. This strongly suggests that the probe light is detecting contributions from magnetized C60, contradicting the authors' assertion that the Kerr signal originates solely from Co. Their response—stating that C60 is transparent at 800 nm—is not convincing because it does not account for the fact that an out-of-equilibrium C60 layer (excited by the pump pulse) could have altered optical properties. The authors should explore this possibility rather than dismissing it outright.

- **Overreliance on the arXiv Manuscript:** A key part of this study's analysis appears to be conducted and justified elsewhere, in a separate arXiv manuscript. This raises concerns about the independence of the current work and whether it stands on its own merit. The authors frequently defer critical discussions to another paper on arXiv, stating that this arxiv paper (1) proves that the FMR stems from Co rather than C60, (2) explains why

Co/C60 and Co/Al have similar coercivity and anisotropy in Figure 1b, and (3) describes quantitative values of the anisotropy and Kittel behavior of the FMR.

If these essential aspects are primarily addressed elsewhere, then what exactly is the novel contribution of the present manuscript? Simply demonstrating that FMR frequency shifts upon laser excitation is not sufficient to warrant publication. Moreover, other key aspects, such as trends in the decay rate and oscillation amplitudes, remain largely unexplored, with the justification that the system is too complex. This significantly weakens the study's impact.

- Overstatement of Technological Relevance: There is excessive use of promotional language throughout the manuscript, with claims of "novelty," "groundbreaking methods," and enabling "next-generation qICT technologies" and a "new paradigm in spin-based information processing." I find the claims very exaggerating; The described technique requires femtosecond optical pulses, operates at cryogenic temperatures, and necessitates high magnetic fields ($\sim 0.5T$) none of which are welcomed in technology. Also, I don't see any "new paradigm" proposed or demonstrated here. Modulating magnetic properties with optical pulses are well-established. The use of a new material platform in this "established" paradigm does not mean there is a "new paradigm".

The paper's focus should be on the fundamental physics of the platform, rather than prematurely marketing it as a technological breakthrough. If the authors wish to emphasize technological relevance, they must provide supporting data on power consumption, device footprint, scalability, and efficiency. Otherwise, the manuscript should be revised to focus on the fundamental physics and allow readers to assess the novelty of the findings.

Other minor comments:

- The figure caption states a 400 nm pump, while the main text refers to 350 nm. This inconsistency needs clarification.
- The authors attribute the step-like behavior in their data to sample degradation. However, this raises concerns about whether such degradation could also affect the pump-probe measurements. A discussion on this possibility is warranted.
- A systematic experiment could involve inserting a conductive layer (e.g., Cu) between C60 and Co. A Cu interlayer would allow charge transfer while blocking interfacial hybridization, providing further insight into the role of C60. The authors should consider such an experiment to strengthen their claims.

Reviewer #1 (Remarks to the Author):

While I acknowledge the presence of some interesting findings, the manuscript in its current form lacks the necessary depth and rigor to be recommended for publication. The core physical mechanisms remain insufficiently substantiated, critical analyses are outsourced to another arXiv paper, and exaggerated claims about technological impact detract from the scientific merit of the study.